# Monitoring induced distributed double-couple sources using Marchenko-based virtual receivers

Joeri Brackenhoff[1], Jan Thorbecke[1], and Kees Wapenaar[1]

[1]Department of Geoscience and Engineering, Delft University of Technology, Stevinweg 1, 2628 CN Delft, The Netherlands

**Correspondence:** Joeri Brackenhoff (j.a.brackenhoff@tudelft.nl)

**Abstract.** We aim to monitor and characterize signals in the subsurface by combining these passive signals with recorded reflection data at the surface of the Earth. To achieve this, we propose a method to create virtual receivers from reflection data using the Marchenko method. By applying homogeneous Green's function retrieval, these virtual receivers are then used to monitor the responses from subsurface sources. We consider monopole point sources with a symmetric source signal, where the full wavefield without artifacts in the subsurface can be obtained. Responses from more complex source mechanisms, such as double-couple sources, can also be used and provide results with comparable quality as the monopole responses. If the source signal is not symmetric in time, our technique that is based on homogeneous Green's function retrieval provides an incomplete signal, with additional artifacts. The duration of these artifacts is limited and they are only present when the source of the signal is located above the virtual receiver. For sources along a fault rupture, this limitation is also present and more severe due to the source activating over a longer period of time. Part of the correct signal is still retrieved, as well as the source location of the signal. These artifacts do not occur in another method which creates virtual sources as well as receivers from reflection data at the surface. This second method can be used to forecast responses to possible future induced seismicity sources (monopoles, double-couple sources and fault ruptures). This method is applied to field data, where similar results to synthetic data are achieved, which shows the potential for the application on real data signals.

## 1 Introduction

Seismic monitoring of processes in the subsurface has been an active field of research for many years. Traditionally, most recording setups are limited to the surface of the Earth, although boreholes can also be utilized. The latter approach is more expensive and complicated, however. In case of monitoring with active sources, the receivers in these recording setups measure valuable reflection data, which provide quantifiable information about processes in the subsurface. Some examples of using this information are monitoring time-shifts in seismic data to predict the velocity-strain relation for a depleting reservoir (Hatchell and Bourne, 2005) and the monitoring of geomechanics in the subsurface by using time-lapse data (Herwanger and Horne, 2009). The responses from passive sources, such as when the signal is caused by an induced earthquake, can be measured as well. These passive measurements are more difficult to process due to the fact that the signal is complex and unknown (McClellan et al., 2018), however, the information content in these induced seismic signals is of great interest. Induced seismicity has had a large impact in countries such as the Netherlands (van Thienen-Visser and Breunese, 2015) and

the USA (Magnani et al., 2017) and there is much discussion about the cause and the effects. To determine the cause of induced seismicity, the source of the signal is of particular interest and consequently, inversions for the source mechanism (Zhang and Eaton, 2018) as well as the location of the source (Eisner et al., 2010) are often performed. These methods can be carried out from surveys that are located at the surface of the Earth or inside boreholes, however, they are limited in accuracy. Ideally, one

would use a dense network of receivers around the source location to directly monitor the wavefield.

Due to practical difficulties and expenses associated with placing a dense network of receivers in the subsurface, the wavefield can generally not be directly measured around the source location of the signal. An alternative to using physical receivers for these measurements is the use of virtual receivers. A virtual receiver is not physically present in the subsurface, rather, it is created through processing of measured signals at the surface. Virtual receivers can be created in a variety of ways. A

mathematical basis for the retrieval of these virtual receivers is the so-called homogeneous Green's function representation. The classical form of this representation was proposed by Porter (1970) and extended for inverse source problems by Porter and Devaney (1982) and for inverse scattering methods by Oristaglio (1989). This representation states that if the responses from two signals are measured on an enclosing recording surface, the response between the two sources of the signals can be retrieved. It forms the basis for seismic interferometry to create virtual sources (Wapenaar et al., 2005) or virtual receivers

(Curtis et al., 2009). All of these approaches require access to the medium from an enclosing surface and introduce artifacts if this requirement is not met. Even though this limitation is well known, for many cases these approaches are still utilized.

A novel approach that can be used when the acquisition surface is not closed is the data-driven 3D Marchenko method. This method can create virtual sources and receivers in the subsurface (Wapenaar et al., 2014; Slob et al., 2014). In order to achieve this, the method requires a reflection response recorded at the surface of the Earth, and an estimation of the first

arrival of the signal from a location in the subsurface to the receiver locations in the measurement array. This first arrival can be estimated from a background velocity model, which requires no detailed information about the subsurface. Through the Marchenko method, the Green's function with a virtual receiver in the subsurface can be retrieved. Using this method, many virtual receivers can be created in the subsurface, which can be used to monitor the wavefield from the virtual receiver locations to the receiver array. To obtain the signal between an induced signal from the subsurface and the virtual receiver

locations, homogeneous Green's function retrieval can be employed, however, as pointed out before, the classical approach would include artifacts due to the open surface of the recording. These artifacts can disturb the interpretation of the signal. An alternative retrieval scheme was developed by Wapenaar et al. (2016), who showed that if a focusing function is used in combination with a Green's function, an open surface can be used for the retrieval instead of an enclosing one, without the artifacts of the classical method when applied to an open surface. A focusing function is a wavefield that is designed to focus at

a location in the subsurface and can be retrieved from reflection data using the Marchenko method (van der Neut et al., 2015). This single-sided representation has been proven to work succesfully on both synthetic data and on field data (Brackenhoff et al., 2019).

Using the single-sided method, two approaches for monitoring induced seismicity can be taken. First, virtual receivers can be used in combination with a virtual source. In this case, all the signals are created from the reflection data using the Marchenko

method. This has the benefit that the virtual source can be created at any location in the subsurface, where one expects induced

seismicity to happen, and that the source signal can be controlled. This is the way that the method has been mostly applied in previous works. Another approach that can be taken is to create virtual receivers using the Marchenko method and to use a real induced seismic source signal instead of a virtual Green's function. This effectively allows for the monitoring of the actual signal in the subsurface, including the source location and mechanism. This could be a boon to induced seismicity monitoring, however, this approach does require some modifications. Induced seismicity often causes more complex source signals that evolve over a period of time and cover an extended area in the subsurface. These rupture planes or fault sources are the main topic of interest.

In this work, we aim to apply the single-sided homogeneous Green's function retrieval on both synthetic and field data for a distribution of virtual double-couple sources. We first apply the method on synthetic data for point sources and show the principles of the representation. We then use the same synthetic data to apply the representation with modifications to the sources originating from a fault plane and show the results that can be achieved. Finally, we also apply the representation on field data for both types of sources.

## 2 Theory

### 2.1 Green's function and focusing function

In this paper, we present several representations for the retrieval of wavefields in the subsurface. First, we review the properties and quantities that are relevant for these representations. To this end, we consider a medium that is acoustic, lossless and inhomogenous with mass density $\rho(\boldsymbol{x})$ and compressibility $\kappa(\boldsymbol{x})$, where $\boldsymbol{x} = (x_1, x_2, x_3)$ indicates the Cartesian coordinate vector. We make use of a Green's function in this medium that obeys the following wave equation:

$$\partial_i(\rho^{-1}\partial_i G) - \kappa\partial_t^2 G = -\delta(\boldsymbol{x} - \boldsymbol{x}_A)\partial_t \delta(t), \tag{1}$$

where $G(\boldsymbol{x}, \boldsymbol{x}_A, t)$ indicates a Green's function that at time $t$ describes the response of the medium at location $\boldsymbol{x}$ due to an unit impulsive point source of volume-injection rate density $\delta(\boldsymbol{x} - \boldsymbol{x}_A)\delta(t)$ at source location $\boldsymbol{x}_A$. $\delta(\cdot)$ is the Dirac delta function, $\partial_t$ the temporal partial differential operator $\frac{\partial}{\partial_t}$ and $\partial_i$ a component of a vector containing the spatial partial differential operators in the three principal directions $\left(\frac{\partial}{\partial x_1}, \frac{\partial}{\partial x_2}, \frac{\partial}{\partial x_3}\right)$. Einstein's summation convention applies to repeated subscripts. The Green's function obeys source-receiver reciprocity, which allows the interchange of the source and receiver position, hence $G(\boldsymbol{x}_B, \boldsymbol{x}_A, t) = G(\boldsymbol{x}_A, \boldsymbol{x}_B, t)$. We impose causality on the Green's function, $G(\boldsymbol{x}, \boldsymbol{x}_A, t) = 0$ for $t < 0$, such that it is forward propagating, originating from the source, and a causal solution to equation (1). A schematic illustration of the Green's function is shown in Figure 1-(a), where several possible raypaths are drawn for a heterogeneous model. This includes the direct arrival, primary reflections and multiple reflections.

We also consider the time-reversed Green's function $G(\boldsymbol{x}, \boldsymbol{x}_A, -t)$, which is the acausal solution to equation (1), where the causality condition implies $G(\boldsymbol{x}, \boldsymbol{x}_A, -t) = 0$ for $t > 0$. Superposition of the causal and acausal Green's function yields the homogeneous Green's function:

$$G_{\mathrm{h}}(\boldsymbol{x}, \boldsymbol{x}_A, t) = G(\boldsymbol{x}, \boldsymbol{x}_A, t) + G(\boldsymbol{x}, \boldsymbol{x}_A, -t), \tag{2}$$

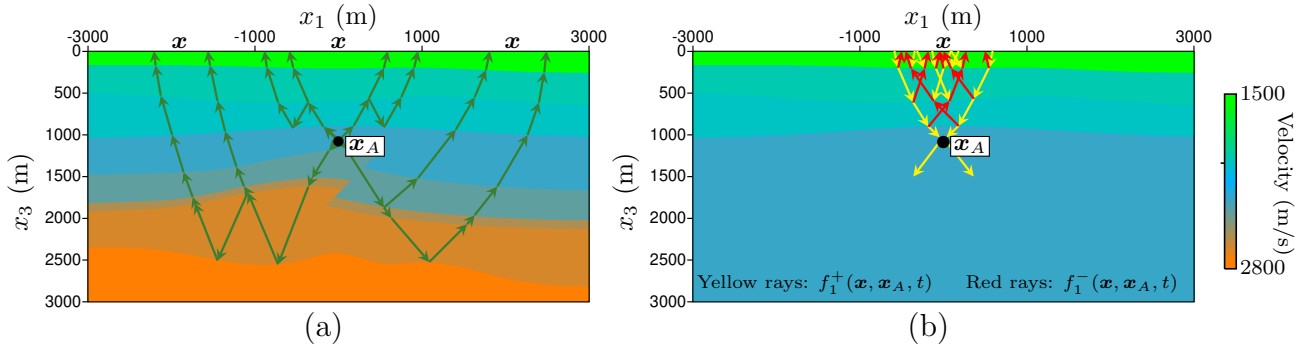

**Figure 1.** (a) Schematic representation of the Green's function $G(\boldsymbol{x}, \boldsymbol{x}_A, t)$, defined in the physical medium, with a source located at $\boldsymbol{x}_A$, which is measured at varying location $\boldsymbol{x}$ at the surface. (b) Schematic representation of the focusing function $f_1(\boldsymbol{x}, \boldsymbol{x}_A, t)$, defined in the truncated medium, where the wavefield propagates from $\boldsymbol{x}$ at the surface to the focal location $\boldsymbol{x}_A$. For both functions, several possible raypaths are drawn. For the focusing function the downgoing waves are marked with yellow arrows and the upgoing waves with red arrows.

where $G_{\mathrm{h}}(\boldsymbol{x}, \boldsymbol{x}_A, t)$ obeys the homogeneous wave equation:

$$\partial_i(\rho^{-1}\partial_i G_{\mathrm{h}}) - \kappa \partial_t^2 G_{\mathrm{h}} = 0. \tag{3}$$

Equation (3) is similar to equation (1), with the exception of the lack of a source singularity on the right hand side of the equation.

Aside from the Green's function, we consider the focusing function $f_1(\boldsymbol{x}, \boldsymbol{x}_A, t)$, which describes a wavefield, at time $t$ and location $\boldsymbol{x}$, that converges to a focal location $\boldsymbol{x}_A$ in the subsurface of a medium that is truncated below the focal location. The focusing function can be decomposed as,

$$f_1(\boldsymbol{x}, \boldsymbol{x}_A, t) = f_1^+(\boldsymbol{x}, \boldsymbol{x}_A, t) + f_1^-(\boldsymbol{x}, \boldsymbol{x}_A, t), \tag{4}$$

where $f_1^+(\boldsymbol{x}, \boldsymbol{x}_A, t)$ denotes the downgoing and $f_1^-(\boldsymbol{x}, \boldsymbol{x}_A, t)$ the upgoing component of the focusing function. A schematic representation of the focusing function can be found in Figure 1-(b). Similar to the Green's function, several possible raypaths are drawn, however, to distinguish the decomposed wavefields, the downgoing focusing function is marked with yellow rays and the upgoing focusing function with red rays. The medium of the focusing function and the Green's function are identical until the focal depth, after which the medium of the focusing function becomes truncated. The physical and truncated medium can be used in reciprocity theorems in order to relate the focusing function to the Green's function, which is shown in section 2 of the supplementary information. For moderately inhomogeneous media, the focusing function and Green's function can be separated from each other in time. The coda of the focusing function resides in the interval between the direct arrival of a related Green's function and its time reversal. The direct arrival of the focusing function coincides with the direct arrival of the time reversed Green's function. This difference in time intervals explains some of the effects that are present in the representations that are used in this paper. Both the focusing function and Green's function can be retrieved for a heterogeneous medium from

the reflection data and an estimate of the direct arrival, through use of the Marchenko method. We will not explain this method in detail in this paper, instead we refer the reader to Wapenaar et al. (2014) for a more detailed overview.

Due to the nature of some equations, we also make use of the frequency domain version of the time domain quantities. To obtain these transformation we make use of the Fourier transform. We define the Fourier transform of a space- and time-dependent function $u(\boldsymbol{x}, t)$ as

$$u(\boldsymbol{x}, \omega) = \int_{-\infty}^{\infty} u(\boldsymbol{x}, t) \exp(i\omega t) \mathrm{d}t, \tag{5}$$

where $u(\boldsymbol{x}, \omega)$ is the Fourier transformed version of $u(\boldsymbol{x}, t)$ in the space-frequency domain, with $\omega$ as the angular frequency and $i$ the imaginary unit. By using equation (5) we obtain the space-frequency domain versions of equation (1), (2), (3) and (4), respectively:

$$\partial_i(\rho^{-1}\partial_i) + \kappa\omega^2 G = i\omega\delta(\boldsymbol{x} - \boldsymbol{x}_A), \tag{6}$$

$$G_{\mathrm{h}}(\boldsymbol{x}, \boldsymbol{x}_A, \omega) = G(\boldsymbol{x}, \boldsymbol{x}_A, \omega) + G^*(\boldsymbol{x}, \boldsymbol{x}_A, \omega) = 2\Re\{G(\boldsymbol{x}, \boldsymbol{x}_A, \omega)\}, \tag{7}$$

$$\partial_i(\rho^{-1}\partial_i G_{\mathrm{h}}) + \kappa\omega^2 G_{\mathrm{h}} = 0, \tag{8}$$

$$f_1(\boldsymbol{x}, \boldsymbol{x}_A, \omega) = f_1^+(\boldsymbol{x}, \boldsymbol{x}_A, \omega) + f_1^-(\boldsymbol{x}, \boldsymbol{x}_A, \omega), \tag{9}$$

where $\Re$ indicates the real part of a complex function.

## 2.2 Homogeneous Green's function representation

The classical homogeneous Green's function representation was originally developed for a configuration where the Green's function was measured on an arbitrarily shaped surface enclosing the medium of interest (Porter, 1970; Porter and Devaney, 1982; Oristaglio, 1989). The representation states that, if the responses from two sources inside the medium are recorded on the surface, the response between the two source locations can be obtained. For seismic recording setups, the measurements are usually only available at the surface of the Earth, meaning that the surface is single-sided instead of closed, which will introduce significant errors into the final result.

In recent years a new representation for homogeneous Green's function retrieval was developed that is designed to work with the single-sided surface, where a focusing function is used together with a Green's function (Wapenaar et al., 2016). Consider the setup in Figure 2, where a heterogeneous medium $\mathbb{V}_A$ is bounded by two horizontal surfaces $\mathbb{S}_0$ and $\mathbb{S}_A$ on two different levels in vertical direction $x_3$. The surfaces extend infinitely in the horizontal directions $x_1$ and $x_2$. The medium above $\mathbb{S}_0$ is homogeneous, with mass density $\rho_0$ and compressibility $\kappa_0$, and the surface itself is non-reflecting, while the medium below $\mathbb{S}_A$ can be heterogeneous. The upper surface $\mathbb{S}_0$ corresponds to the surface where the receiver locations $\boldsymbol{x}$ of focusing functions and Green's functions are available. In this scenario, we assume that we have three functions available at the upper surface, a Green's function $G(\boldsymbol{x}, \boldsymbol{x}_B^{(1)}, \omega)$, that has a source location $\boldsymbol{x}_B^{(1)}$ below $\mathbb{S}_A$, a Green's function $G(\boldsymbol{x}, \boldsymbol{x}_B^{(2)}, \omega)$, that has a source

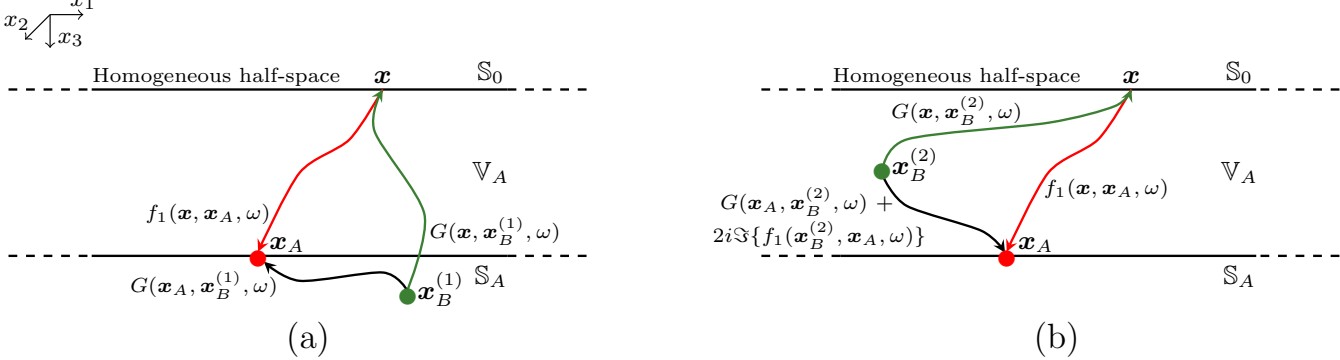

**Figure 2.** Setup for the single-sided Green's function representation for (a) a case where the source of the Green's function is located below the focal location and (b) a case where the source of the Green's function is located above the focal location. The rays in this figure indicate full Green's functions and focusing functions, including multiple scattering.

location $\boldsymbol{x}_B^{(2)}$ inside medium $\mathbb{V}_A$ and a focusing function $f_1(\boldsymbol{x}, \boldsymbol{x}_A, \omega)$, that has a focal location $\boldsymbol{x}_A$, located at the depth of $\mathbb{S}_A$. The available functions can be used to obtain the response between two locations. To this end, we use the representation given by equation (35) of the supplementary information (for the derivation see section 2.3 of the supplementary material),

$$G(\boldsymbol{x}_A, \boldsymbol{x}_B, \omega) + \chi(\boldsymbol{x}_B) 2i\Im\{f_1(\boldsymbol{x}_B, \boldsymbol{x}_A, \omega)\} = \int_{\mathbb{S}_0} \frac{2}{i\omega\rho_0} G(\boldsymbol{x}, \boldsymbol{x}_B, \omega)\partial_3(f_1^+(\boldsymbol{x}, \boldsymbol{x}_A, \omega) - \{f_1^-(\boldsymbol{x}, \boldsymbol{x}_A, \omega)\}^*)\mathrm{d}\boldsymbol{x}, \qquad (10)$$

where $\Im$ is the imaginary part of a complex function and $\chi(\boldsymbol{x}_B)$ is the characteristic function,

$$\chi(\boldsymbol{x}_B) = \begin{cases} 1, & \text{for } \boldsymbol{x}_B \text{ in } \mathbb{V}_A, \\ \frac{1}{2}, & \text{for } \boldsymbol{x}_B \text{ on } \mathbb{S} = \mathbb{S}_0 \cup \mathbb{S}_A, \\ 0, & \text{for } \boldsymbol{x}_B \text{ outside } \mathbb{V}_A \cup \mathbb{S}. \end{cases} \qquad (11)$$

This representation states that, by applying the focusing function components to a Green's function at the upper surface, the Green's function between the focal location $\boldsymbol{x}_A$ of the focusing function and the source location $\boldsymbol{x}_B$ of the Green's function can be obtained. The focal location will become the receiver of this new Green's function, and the source location of the original Green's function on the right hand side of equation (10) will become the source location of the new Green's function. However, contributions from the imaginary part of the focusing function between the source and receiver locations are present if the source location is located inside the medium $\mathbb{V}_A$, as is the case if the Green's function from Figure 2-(b) with source location $\boldsymbol{x}_B^{(2)}$ is used. Because they are related to a focusing function, these artifacts will be present between the direct arrival of the Green's function and its time reversal. In this case, the source location is present above the focal location. These contributions vanish if the source location is present outside $\mathbb{V}_A$, in other words if it is located below the focal location, such as when the Green's function from Figure 2-(a) with source location $\boldsymbol{x}_B^{(1)}$ is used. This would mean that, without knowledge of $\Im\{f_1(\boldsymbol{x}_B, \boldsymbol{x}_A, \omega)\}$, we are limited in the correct application of the representation. To overcome this limitation, we substitute

equation (10) into the right hand side of equation (7) to create the single-sided homogeneous Green's function representation:

$$G_{\mathrm{h}}(\boldsymbol{x}_A,\boldsymbol{x}_B,\omega) = 4\Re \int\limits_{\mathbb{S}_0} \frac{1}{i\omega\rho_0} G(\boldsymbol{x},\boldsymbol{x}_B,\omega)\partial_3\big(f_1^+(\boldsymbol{x},\boldsymbol{x}_A,\omega) - \{f_1^-(\boldsymbol{x},\boldsymbol{x}_A,\omega)\}^*\big)\mathrm{d}\boldsymbol{x}, \tag{12}$$

5   which corresponds to equation (33) from our companion paper (Wapenaar et al., 2019). The additional contributions have vanished from this representation and the homogeneous Green's function will be obtained when it is evaluated, instead of the causal Green's function.

## 2.3   Virtual sources and receivers

10   Generally, the focusing function and Green's function are not directly available. These functions can be obtained through the use of the Marchenko method (Broggini et al., 2012; Wapenaar et al., 2014; van der Neut et al., 2015), which is a data-driven method that requires only reflection data at the surface of the Earth and an estimation of the first arrival of the wavefield at the location of interest inside the medium. The method handles the primaries of the reflection data in the same way as conventional methods, however, unlike those methods, the Marchenko method can also correctly handle the multiples in the

15   data. The first arrival can be estimated through the use of a macro-velocity model. The method cannot handle attenuation on the reflection data and ignores evanescent waves. On field data, the data requires additional processing to account for these and other requirements. The Marchenko method has been applied succesfully on both synthetic and field data, for examples see Ravasi et al. (2016), Staring et al. (2018) and Brackenhoff et al. (2019) .

The method can be used in the homogeneous Green's function retrieval scheme in two ways, which are schematically shown

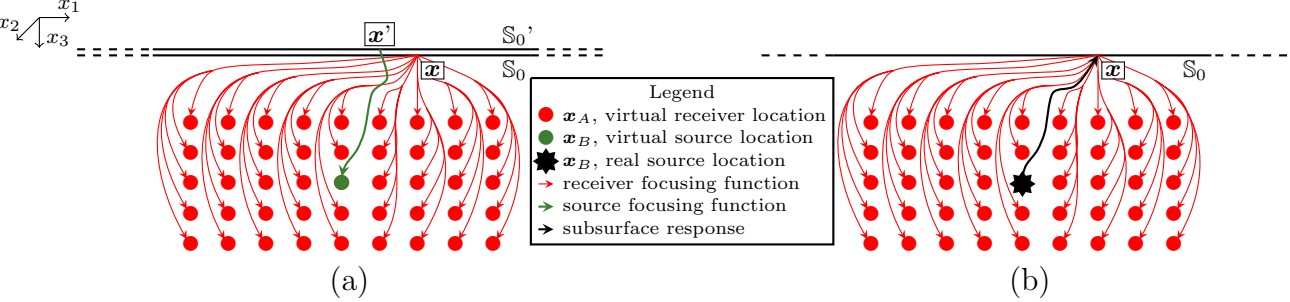

**Figure 3.** Schematic setup for (a) the two step process and (b) the one step process for retrieving the homogeneous Green's function in the subsurface. The red and green arrows show the focusing functions that are used to respectively create the virtual receiver and virtual source location. The red and green dots show the locations for the virtual receiver and virtual source, respectively. The black star indicates the source location of a real subsurface response, indicated with a black arrow, that is measured at the surface $\mathbb{S}_0$ on the same receiver location $\boldsymbol{x}$ as the focusing and Green's functions. $\mathbb{S}_0$' is a surface located just above $\mathbb{S}_0$ on which the source locations $\boldsymbol{x}$' of the reflection response $p(\boldsymbol{x},\boldsymbol{x}',\omega)$ are located. The rays in this figure indicate full Green's functions and focusing functions, including multiple scattering.

in Figure 3. The first approach is a two-step process, as shown in Figure 3-(a), where both the source and receiver of the

homogeneous Green's function are obtained by redatuming them from the reflection response. This type of source-receiver

redatuming is discussed in section 3.4 of our companion paper by Wapenaar et al. (2019). First, we consider the fact that the data that we use in the field is bandlimited and therefore a source signal $s(t)$ is convolved with the Green's function, which changes its phase and amplitude:

$$p(\boldsymbol{x}, \boldsymbol{x}_B, t) = \int_{-\infty}^{\infty} G(\boldsymbol{x}, \boldsymbol{x}_B, t - t') s(t') \mathrm{d}t', \tag{13a}$$

$$p(\boldsymbol{x}, \boldsymbol{x}_B, \omega) = G(\boldsymbol{x}, \boldsymbol{x}_B, \omega) s(\omega), \tag{13b}$$

where $p(\boldsymbol{x}, \boldsymbol{x}_B, t)$ is a pressure wavefield in the medium and $s(\omega)$ is the Fourier transform of the source signal. For the first step, we introduce a second surface $\mathbb{S}_0'$ that is located just above $\mathbb{S}_0$ and assume that a reflection response $p(\boldsymbol{x}, \boldsymbol{x}', \omega)$ of the medium has been measured, where $\boldsymbol{x}'$ is the source location on the surface $\mathbb{S}_0'$. The reflection response is used to create a virtual source location in the subsurface. To this end, we utilize a modification of equation (12), and use equation (13b) to create an equivalent version for pressure wavefields, which is the same as equation (41) of our companion paper:

$$p(\boldsymbol{x}, \boldsymbol{x}_B, \omega) + p^*(\boldsymbol{x}, \boldsymbol{x}_B, \omega) = 4\Re \int_{\mathbb{S}_0'} \frac{1}{i\omega\rho_0} p(\boldsymbol{x}, \boldsymbol{x}', \omega) \partial_3 \left( f_1^+(\boldsymbol{x}', \boldsymbol{x}_B, \omega) - \{f_1^-(\boldsymbol{x}', \boldsymbol{x}_B, \omega)\}^* \right) \mathrm{d}\boldsymbol{x}'. \tag{14}$$

In equation (14), we assume that the source spectrum is strictly real-valued. The focusing function $f_1(\boldsymbol{x}', \boldsymbol{x}_A, \omega)$ is obtained through use of the Marchenko method and employed in equation (14) to create a wavefield with a virtual source location, which is indicated by the green line in Figure 3-(a). This function will be used to create a source location for the wavefield retrieved through the homogeneous Green's function representation. This source is called a virtual source because it is not physically

present in the subsurface.

In the second step of the process, using the Marchenko method, many focusing functions are created for focal points at varying locations in the medium, that serve as the virtual receiver locations for the retrieved wavefield. This is indicated by the red dots and arrows in Figure 3-(a). Similarly to the virtual source, these are called virtual receivers, again, because they are not physically present in the medium. We use these focusing functions in equation (10), which we modify using equation equation

(13b) as follows

$$p(\boldsymbol{x}_A, \boldsymbol{x}_B, \omega) + \chi(\boldsymbol{x}_B) 2is(\omega) \Im\{f_1(\boldsymbol{x}_B, \boldsymbol{x}_A, \omega)\} = \int_{\mathbb{S}_0} \frac{2}{i\omega\rho_0} p(\boldsymbol{x}, \boldsymbol{x}_B, \omega) \partial_3 (f_1^+(\boldsymbol{x}, \boldsymbol{x}_A, \omega) - \{f_1^-(\boldsymbol{x}, \boldsymbol{x}_A, \omega)\}^*) \mathrm{d}\boldsymbol{x}. \tag{15}$$

In this representation, we make use of the wavefield $p(\boldsymbol{x}, \boldsymbol{x}_B, \omega)$ with the virtual source location that we obtained in the first step. The acausal part of the left hand side of the time-domain version of equation (14) can be removed easily by applying causality through the use of a Heaviside function. Since we assumed $s(\omega)$ to be real-valued, substitution of equation (15) into equation (7) yields,

$$p_{\mathrm{h}}(\boldsymbol{x}_A, \boldsymbol{x}_B, \omega) = 4\Re \int_{\mathbb{S}_0} \frac{1}{i\omega\rho_0} p(\boldsymbol{x}, \boldsymbol{x}_B, \omega) \partial_3 \left( f_1^+(\boldsymbol{x}, \boldsymbol{x}_A, \omega) - \{f_1^-(\boldsymbol{x}, \boldsymbol{x}_A, \omega)\}^* \right) \mathrm{d}\boldsymbol{x}, \tag{16}$$

where $p_{\mathrm{h}}(\boldsymbol{x}_A, \boldsymbol{x}_B, \omega) = p(\boldsymbol{x}_A, \boldsymbol{x}_B, \omega) + p^*(\boldsymbol{x}_A, \boldsymbol{x}_B, \omega)$. This is a similar representation to equation (39) for modified back propagation from our companion paper by Wapenaar et al. (2019).

The second way we can use the Marchenko method in the application of homogeneous Green's function retrieval is a one-step process, where the Marchenko method is only used to retrieve focusing functions to create virtual receivers. This is shown in Figure 3-(b). Here, no virtual source is created from the reflection data using equation (14), rather the actual response from a real source inside the medium is used, which is illustrated by the black star and arrow in Figure 3-(b). The response that is monitored is used as $p(\boldsymbol{x}, \boldsymbol{x}_B, \omega)$ in equation (15). It can not generally be used in equation (16), however. If the source spectrum

of the response is not strictly real-valued, the signal is not symmetric in time, because $s(\omega) \neq s^*(\omega)$, and therefore there will be a phase difference between the causal and acausal wavefield, making the superposition of the signal with its time-reverse incorrect. Assuming that through processing of the signal, the type of wavelet that is applied to the data can be controlled, symmetry of the source signal can be ensured by using zero-phase wavelets. When this condition is fulfilled, equation (16) can be used for the subsurface response. Monitoring real source signals is the eventual goal of this approach, such as for the case

of induced seismicity. The boon of this method is that aside from the measured signal, no information about the source of the data is required. There are limitations to this approach as well, most pressing that to evaluate the integral, the signal needs to be recorded on the same receiver array that was used to record the reflection data.

## 2.4 Modifications for realistic induced seismicity sources

### 2.4.1 Double-couple point sources

For the case of induced seismicity, the source signal can be more complex than just a single monopole point source. To include the mechanics for induced earthquakes more accurately, the double-couple source mechanism can be included in the representation. The double-couple source mechanism is accepted as representative for an earthquake response if the wavelength of the signal is at least of the same dimension as the size of the fault that originated the earthquake (Aki and Richards, 2002).

It can be implemented through the use of a moment tensor, which is useful for the case of finite-difference modeling (Li et al., 2014). The response of a monopole source and double-couple source for a homogeneous medium is shown in Figure 4, along with their radiation patterns in the center. While the monopole source response has an uniform amplitude along the wavefront, the double-couple source response has a varying amplitude and polarity along the wavefront, due to the variation in the radiation pattern. Consequently, the orientation of the double-couple source affects the source signal, which is visible in the

Figure 4-(b), while the orientation of the monopole source does not matter. Hence, the orientation of the fault is crucial to the characteristics of the double-couple source signal. To include this orientation in the representation, we introduce the operator $\mathfrak{D}_B^{\theta}$, which acts on the wavefield and creates the double-couple source orientation from the monopole source signature. This operator is defined as

$$\mathfrak{D}_B^{\theta} = (\theta_i^{\parallel} + \theta_i^{\perp})\partial_{i,B}, \tag{17}$$

5  where $\partial_{i,B}$ is a component of the vector containing the partial derivatives acting on the monopole signal originating from source location $\boldsymbol{x}_B$, that turns it into a double-couple source mechanism, $\theta_i^{\parallel}$ is a component of the unit vector that orients one couple of the signal parallel to the fault plane and $\theta_i^{\perp}$ is a component of the vector that orients the other couple perpendicular

to the fault plane. The operator can be applied to equation (15):

$$\mathfrak{D}_B^\theta\{p(\boldsymbol{x}_A,\boldsymbol{x}_B,\omega)\}+\mathfrak{D}_B^\theta\{\chi(\boldsymbol{x}_B)2is(\omega)\Im\{f_1(\boldsymbol{x}_B,\boldsymbol{x}_A,\omega)\}\}=$$
$$\int_{\mathbb{S}_0}\frac{2}{i\omega\rho_0}\mathfrak{D}_B^\theta\{p(\boldsymbol{x},\boldsymbol{x}_B,\omega)\}\partial_3(f_1^+(\boldsymbol{x},\boldsymbol{x}_A,\omega)-\{f_1^-(\boldsymbol{x},\boldsymbol{x}_A,\omega)\}^*)\mathrm{d}\boldsymbol{x}, \tag{18}$$

10   and assuming that the source signal is symmetric in time, the operator is also applied to equation (16)

$$\mathfrak{D}_B^\theta\{p_\mathrm{h}(\boldsymbol{x}_A,\boldsymbol{x}_B,\omega)\}=4\Re\int_{\mathbb{S}_0}\frac{1}{i\omega\rho_0}\mathfrak{D}_B^\theta\{p(\boldsymbol{x},\boldsymbol{x}_B,\omega)\}\partial_3\big(f_1^+(\boldsymbol{x},\boldsymbol{x}_A,\omega)-\{f_1^-(\boldsymbol{x},\boldsymbol{x}_A,\omega)\}^*\big)\mathrm{d}\boldsymbol{x}. \tag{19}$$

In these two equations, the operator can be freely applied to both sides, because the integral is not evaluated over the source locations. Consequently, if the wavefield response used as a source for the homogeneous wavefield has a double-couple signature, the homogeneous wavefield will also have a double-couple signature. Note that the operator does not operate on the focusing functions, hence we can use the monopole responses for these signals.

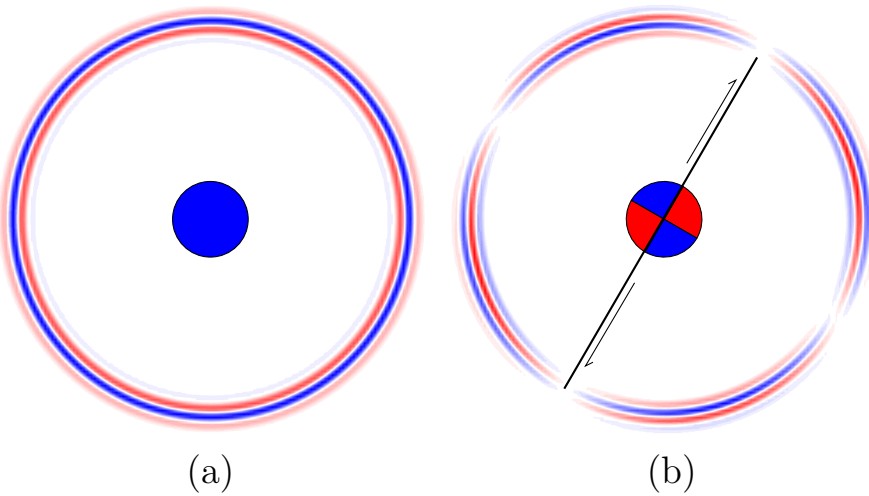

(a)       (b)

**Figure 4.** Comparison of the wavefields caused by (a) a monopole point source and (b) a double-couple point source tilted at an angle of 30 degrees. For both types of sources the radiation pattern of the source is shown in the center. The wavefields have been convolved with a 30 Hz Ricker wavelet.

### 2.4.2 Double-couple sources along extended faults

In case of induced seismicity, the fault or rupture plane that triggers the signal can be larger than the wavelength of the signal.
5   In this case, the double-couple point source is no longer a valid approximation for the source of the signal. Studies of induced faults suggest that the signal develops over the fault during an extended period of time (Buijze et al., 2017). To approximate this type of source, a superposition of many point sources can be utilized. The total signal of the resulting superposition can be

written as the superposition of the individual signals,

$$P(\boldsymbol{x}_A,\omega) = \sum_{k=1}^{N} \mathfrak{D}_B^{\theta,(k)}\{p(\boldsymbol{x}_A,\boldsymbol{x}_B^{(k)},\omega)\} = \sum_{k=1}^{N} \mathfrak{D}_B^{\theta,(k)}\{G(\boldsymbol{x}_A,\boldsymbol{x}_B^{(k)},\omega)s^{(k)}(\omega)\}, \tag{20}$$

where the superscript $k$, indicates the number of the source location $\boldsymbol{x}_B^{(k)}$ that has the source spectrum $s^{(k)}(\omega)$. The different source spectra include a linear phase term that determines the time at which the signal is triggered along the fault plane. $P(\boldsymbol{x}_A,\omega)$ can be created in two different ways, similar as before.

First, we consider the two-step process, where both the source and receiver are virtual. In this case, every source location can be treated separately to retrieve the homogeneous wavefield, and the superposition can be done after each signal has been retrieved through equation (19) and then shifted over $t^{(k)}$,

$$P(\boldsymbol{x}_A,t) = \sum_{k=1}^{N} H(t-t^{(k)})\mathfrak{D}_B^{\theta,(k)}\{p_{\mathrm{h}}(\boldsymbol{x}_A,\boldsymbol{x}_B^{(k)},t-t^{(k)})\}, \tag{21}$$

where $H$ is the Heaviside step function and $t^{(k)}$ is the time at which point the $k$-th signal originates on the fault. The Heaviside in equation (21) selects the shifted causal signal from the shifted homogeneous (two-sided) signal before the superposition takes place, which is required to construct the correct signal. If the shifted homogeneous signals would be used instead, the shifted acausal part of later signals would overlap with the causal part of signals that originated earlier. Through use of equation (21) the correct signal can be retrieved.

In case the source signal is measured rather than virtually created, the same approach cannot be taken. This signal is by definition measured after superposition, therefore each point source cannot be evaluated seperately. To represent this, equation (18) is adjusted to take the implicit superposition into account, according to

$$P(\boldsymbol{x}_A,\omega) + \sum_{k=1}^{N} \mathfrak{D}_B^{\theta,(k)}\{\chi(\boldsymbol{x}_B^{(k)})2is^{(k)}(\omega)\Im\{f_1(\boldsymbol{x}_B^{(k)},\boldsymbol{x}_A,\omega)\}\} =$$

$$\int_{\mathbb{S}_0} \frac{2}{i\omega\rho_0}P(\boldsymbol{x},\omega)\partial_3(f_1^+(\boldsymbol{x},\boldsymbol{x}_A,\omega) - \{f_1^-(\boldsymbol{x},\boldsymbol{x}_A,\omega)\}^*)\mathrm{d}\boldsymbol{x} =$$

$$\int_{\mathbb{S}_0} \frac{2}{i\omega\rho_0}\sum_{k=1}^{N} \mathfrak{D}_B^{\theta,(k)}\{p(\boldsymbol{x},\boldsymbol{x}_B^{(k)},\omega)\}\partial_3(f_1^+(\boldsymbol{x},\boldsymbol{x}_A,\omega) - \{f_1^-(\boldsymbol{x},\boldsymbol{x}_A,\omega)\}^*)\mathrm{d}\boldsymbol{x}. \tag{22}$$

In this scenario, the sum is inside the integral and the entire signal is superposed before the focusing function is applied to it. This also results in a superposition of contributions of the focusing function between the virtual receiver location and the fault plane (i.e., the second term on the left-hand side). Substituting equation (22) into equation (7) will not lead to a cancellation of the focusing function on the left-hand side, as the wavefield does not have a symmetric source signal, due to the time differences between all the sources. As such, equation (22) is the endpoint and we will not obtain a homogeneous wavefield, but rather a signal between the source and virtual receiver plus additional artifacts caused by the focusing funtion between the virtual receiver and the fault plane. Similar to the single source, each set of artifacts maps in between the shifted direct arrival of the wavefield and its time-reversal. Due to the different shift of each signal, the artifacts overlap with the shifted causal

and acausal parts of other signals and cannot be easily separated. However, because of the limited duration of the artifacts, the signal at later times will be free from these artifacts. Additionally, due to the nature of the characteristic function, the artifacts also vanish when the source location $x_B^{(k)}$ is outside the volume $\mathbb{V}_A$. In other words, if the virtual receiver location $x_A$ is above the shallowest source location, the correct signal can be retrieved for this virtual receiver.

## 3 Results

### 3.1 Numerical results

#### 3.1.1 Monopole and double-couple point sources

To demonstrate the different approaches to homogeneous Green's function retrieval, we apply the methods first on synthetic data. Figure 5-(a) shows a density model and Figure 5-(b) shows the accompying P-wave velocity model. The model contains an area of faulting in the center of the model, which is highlighted with a black dashed line. To create the required reflection data, the model is used in a finite-difference modeling code for wavefield modeling (Thorbecke and Draganov, 2011). An example of an acoustic common-source record from the center of the model is shown in Figure 5-(c). This type of common-source records and a smoothed version of the velocity model in Figure 5-(b), are the only input that we will use for our applications. To retrieve the required Green's functions and focusing functions with the Marchenko method, we model the first arrival from a point in the subsurface to the surface of the medium using the smooth velocity model and a homogeneous density model. This first arrival is then used to initiate the Marchenko method to retrieve focusing functions and a Green's function from the reflection response at the surface (i.e., from the common source records). The scheme that we use is based on the Marchenko code created by Thorbecke et al. (2017). This is a code for an acoustic wavefield Marchenko method, excluding free-surface multiples in the reflection data. Free-surface multiples could be included in the scheme as was shown by Singh et al. (2015), but this beyond the scope of the current paper.

Figure 6 shows the results of the homogeneous Green's function retrieval. All snapshots show the same area in the subsurface, which is denoted by the white box in Figures 5-(a) and (b). Note that the box does not show the true aspect ratio of the area, however, the snapshots in Figure 6 do. Each pixel in the image is a receiver location and the source location for all images is exactly the same. The columns show snapshots of the wavefield in the subsurface at four different points in time, 0, 150, 300 and 450 ms. Each row corresponds to a specific way the wavefield in the subsurface was constructed. In the first row, the source and the receivers of the wavefield are placed inside the model and the wavefield is directly modeled. This is the benchmark that the other results will be compared to. All snapshots contain an overlay of black dashed lines, which indicate the locations of geological layer interfaces. As can be seen in the figure, the wavefield of the modeling scatters at these lines.

The Marchenko based approach is an improvement over classical methods as was shown by Brackenhoff et al. (2019), because of the focusing functions that are utilized. To demonstrate this, we first consider a more conventional approach, namely the classical back propagation method from section 2.4 of our companion paper by Wapenaar et al. (2019), from which we use

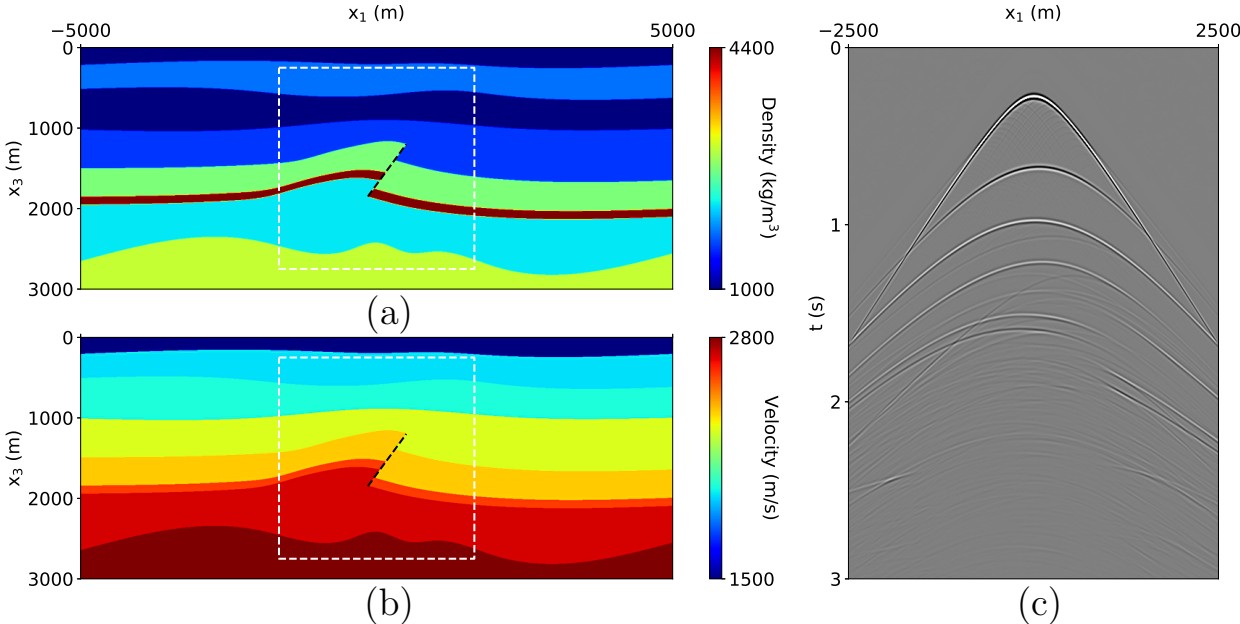

**Figure 5.** (a) Density in $\frac{kg}{m^3}$ and (b) P-wave velocity in $\frac{m}{s}$ of the numerical model used to create reflection data. The white box denotes the area of interest for the purpose of homogeneous Green's function retrieval. The black dashed line indicates a fault plane. (c) Common-source record, created using the model data in (a) and (b), with the source at the top center of the model, using a finite-difference modeling code and convolved with a 30 Hz Ricker wavelet.

equation (23):

$$p^-(\boldsymbol{x}_A,\boldsymbol{x}_B,\omega) \approx \int\limits_{\mathbb{S}_0} \frac{2}{i\omega\rho_0} p^-(\boldsymbol{x},\boldsymbol{x}_B,\omega)\partial_3 G_d^*(\boldsymbol{x},\boldsymbol{x}_A,\omega)\mathrm{d}\boldsymbol{x}, \tag{23}$$

10    where $p^-$ is the upgoing component of the pressure wavefield at $\mathbb{S}_0$ and $G_d^*(\boldsymbol{x},\boldsymbol{x}_A,\omega)$ is the time-reversed first arrival of the Green's function and is the same first arrival that is used as the initial estimation of the focusing function that is used in the Marchenko method. For more information about the method, we refer the reader to our companion paper. Here, we demonstrate the issues with this approach, which can be seen in Figures 6-(e)-(h). The primary upgoing wavefield can be recovered using this method, however, the downgoing wavefield is missing and strong artifacts are present. This is due to the fact that the multiples and the downgoing wavefield are not taken into account properly using the back propagation method. To make a more detailed comparison between the result of this method and the modeling, we extract the measurements from two receiver locations. These locations are indicated in Figure 6-(a), where the red dot is a receiver location above the source location and

the blue dot a receiver location below the source location. Parts of these measurements are displayed in Figure 7, where the left column corresponds to the red dot and the right column to the blue dot. The results in the rows of Figure 7 correspond to the

results of the rows in Figure 6. However, the normalized amplitudes of the traces are used instead of the exact amplitudes. This is done because the first arrivals that were used for the Marchenko method and back propagation were retrieved in a smooth velocity medium without any density information, which is realistic, considering the availibility of data in the field. Because of these limitations the absolute amplitude of the first arrival will be incorrect and while this has no effect on the relative amplitude, it does cause an incorrect overall scaling on the final retrieved wavefield. However, we can still use the normalized traces to analyze the events that are retrieved with the correct relative amplitude. The trace in Figure 7-(c), located above the virtual source, shows that while some of the correct events are retrieved, a large amount of desired events are missing. These problems are more severe for the receiver below the source location. In Figure 7-(d), physical events are missing and there are artifacts present all over the trace. The classical back propagation method lacks a great deal of accuracy.

The third row of Figure 6 shows the result of Green's function retrieval using the method described by equation (15). The Green's function and focusing functions that are required for this method are retrieved using the Marchenko method. This means that all the receivers and the source are virtual. When the result is compared to the benchmark, it is clear that there are some issues. The wavefield below the source location, as indicated by the yellow dashed line, contains numerous artifacts and the downgoing direct arrival of the wavefield is missing, however, the coda of the wavefield is present both above and below the source location, which is a significant improvement over the back propagation. The remaining errors below the source location are caused by the fact that the focusing function between the virtual source and receiver is present and the lack of compensation for these contributions cause artifacts in the final result. When the virtual receivers are located above the virtual source location, the wavefield is comparable to the benchmark and the direct arrival is present. When the trace in Figure 7-(a) is compared to (e), the arrival times of the events match and there are no artifacts present, however there is a mismatch in amplitude. This is due to transmission losses in the reflection response, that the Marchenko method in its current form does not compensate. These effects have been partially compensated for through use of the method discussed by Brackenhoff (2016), although not all the effects have been removed. Also, we expect some numerical issues due to the fact that the modeling and the retrieval of the data are two fundamentally different approaches and the data are discretized. The modeling of the first arrival in the smooth model does not only affect their amplitudes, also the arrival times will shift slightly. Due to this slight shift the sampling points of the modeling and the retrieved wavefield may not match exactly. We ensure that the wavelet is zero-phase for the modeling and the Marchenko method to fulfill the symmetric source signal requirement for the homogeneous Green's function representation. When the receiver location below the source is considered in Figure 7-(f), the results are less accurate. The trace of the modeling contains no signal before the first arrival, whereas the trace for the Green's function retrieval contains numerous events and is lacking the first arrival. The coda of the traces shows a match that is comparable to the receiver location above the source. The arrival times of the events show a good match, while the amplitudes show errors. Because this receiver is located deeper inside the model, the transmission effects are stronger and therefore the error is larger.

Next, the homogeneous Green's function retrieval using equation (16) is considered. The input for this approach is exactly the same as the one used for the previous approach using equation (15), however, this time, we expect to retrieve the correct result. Looking at Figures 6-(m)-(p), the result more closely matches the result of the benchmark. The improvement over the previous result for the deeper virtual receivers is clear. For some of the deeper receivers, part of the wavefield is still not completely

present, however. This is the part of the wavefield that has a steep angle. The reason for this missing part is that the reflection response at the surface does not contain the reflections corresponding to the angles at larger depths, as they travel outside the aperture of the recording survey. Therefore, these steep angles cannot be reconstructed. As can be seen when the trace from

Figure 7-(e) is compared to (g), the result of the two approaches is exactly the same if the virtual receiver is located above the source. The improvement is noticeable when the receiver is located below the source. Figure 7-(h) does contain the first arrival and lacks any signal before this arrival, and therefore shows a better match to Figure 7-(b). While the amplitude mismatch is still present, the arrival times of the events match and no artifacts are present. This also shows that the coda of Figure 7-(f) is correctly retrieved. We have indicated the moment that the correct coda is retrieved with a yellow line in this figure.

To make a more careful comparison between the modeled wavefield and the wavefields retrieved from the reflection data, we plotted the traces from Figures 7-(a)-(h) together in Figure 8, where the left column shows the result for the traces above and the right column shows the result for the traces below the virtual source location. Each subplot contains the modeled response with an overlay of one of the retrieval methods. The back propagation method shows very large errors for both receiver locations as can be seen in Figures 8-(a) and (b). Strong physical events are missing and artifacts are present on both traces. When

comparing the results in Figure 8-(c), the match of the events between the modeled wavefield and the retrieved wavefield is not perfect. As mentioned before, this is due to the influence of the smooth model and numerical effects that occur. A similar match can be seen in Figure 8-(e). The retrieval of the Green's function with the artifacts below the source location, which is displayed in Figure 8-(d), shows the errors at early time, however, also demonstrates that the events in the coda are well captured. This error is not present in case of the homogeneous Green's function retrieval as shown in Figure 8-(f). These results show that

the approach using the Marchenko method is capable of retrieving the relative amplitudes of the events and can retrieve arrival times that are very close to the actual arrival times, even if a smooth velocity model is used.

Finally, we consider the situation where the source mechanism is more complex, through the use of a double-couple signature. The retrieval in this case corresponds to the approach in equation (19), using a virtual source. The double-couple is an elastic mechanism, however, as we only require the first arrival to initiate the Marchenko method, the coda of the wavefield is not of

interest. The S-wave velocity used for the modeling of the first arrival is set to $500\frac{\mathrm{m}}{\mathrm{s}}$, to ensure that all the S-wave events arrive after the first P-wave arrival. We incline the double-couple source at an angle of 45 degrees and use it to model the first arrival, which is used to initiate the Marchenko method to retrieve the wavefield response for the virtual source location. The focusing functions remain the same as the ones we used for the previous approaches in Figures 6-(e)-(h). The result of this retrieval is shown in Figures 6-(q)-(t). As equation (19) states, because the Green's function contains a double-couple signature, the homogeneous Green's function contains the same signature, both in the direct arrival and in the coda of the wavefield. The double-couple signature affects the amplitude of the wavefield depending on the angle of the wavefront, however, the arrival

5    times are similar to those when a monopole virtual source is used. This becomes clear when the traces from Figures 7-(i)-(j) are considered. The arrival times for the events are similar to the previous result, however, there are apparent amplitude and phase differences, caused by the different types of source signature. Due to these differences, we have not included these traces in Figure 8, as a direct comparison between the events cannot be made. The result shows that the double-couple signature can be succesfully integrated in the Marchenko method.

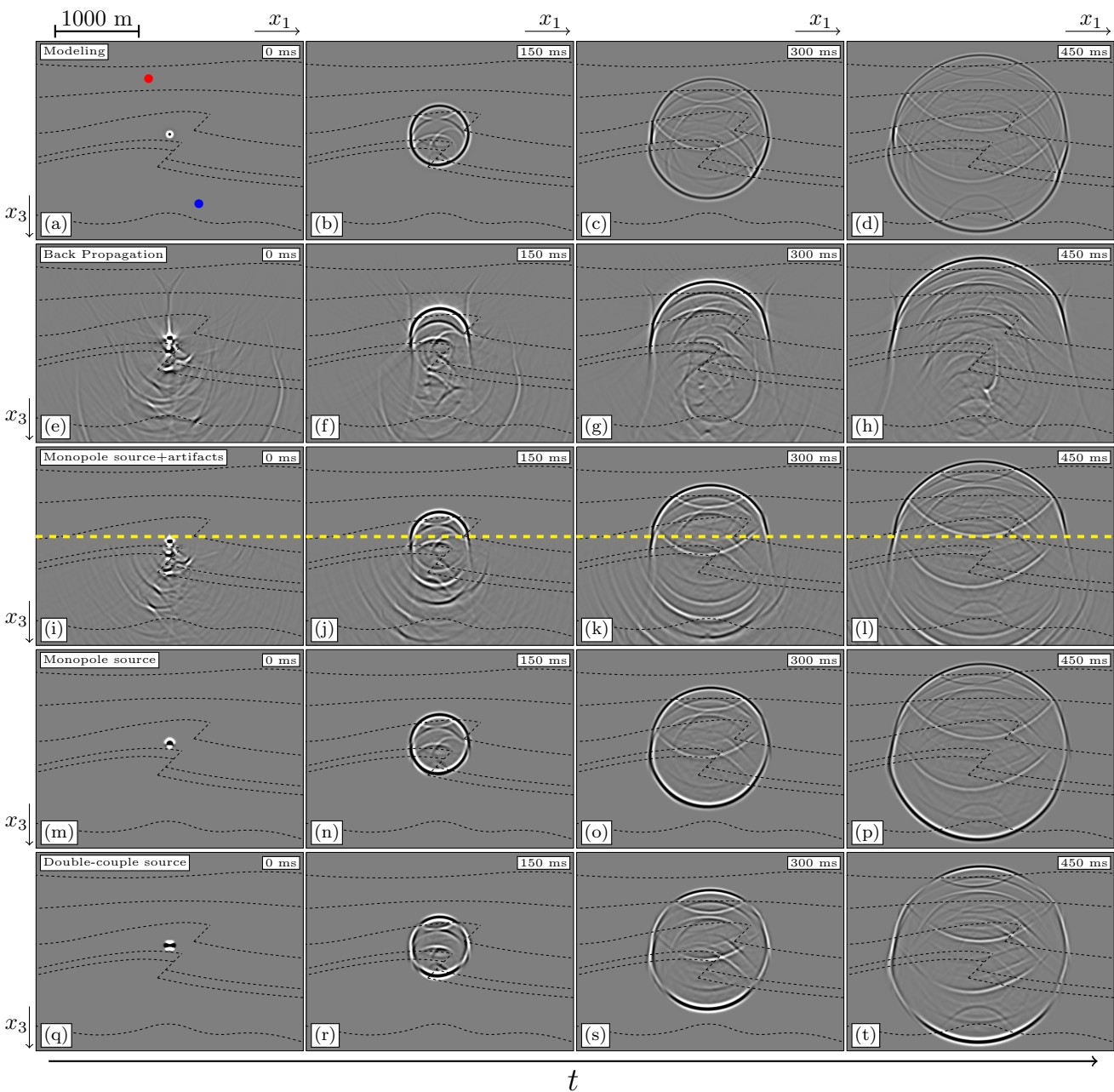

**Figure 6.** Snapshots of the wavefield inside the white box in Figure 5 for point sources. (a)-(d) Directly modeled wavefield using the exact model from Figures 5-(a) and (b). (e)-(h) Back-propagated wavefield obtained using equation (23). (i)-(l) Wavefield in the subsurface, retrieved for virtual receivers and a virtual source using equation (15). The yellow line indicates the border between the area below and above the virtual source. (m)-(p) Similar as (i)-(l), for the homogeneous wavefield using equation (16). (q)-(t) Similar as (m)-(p), using equation (19) and a double couple signature inclined at an angle of 45 degrees. All wavefields have been convolved with a 30 Hz Ricker wavelet. The red and blue dot indicate the locations of the traces in Figure 7. The black dashed lines indicate the locations of geological layer interfaces.

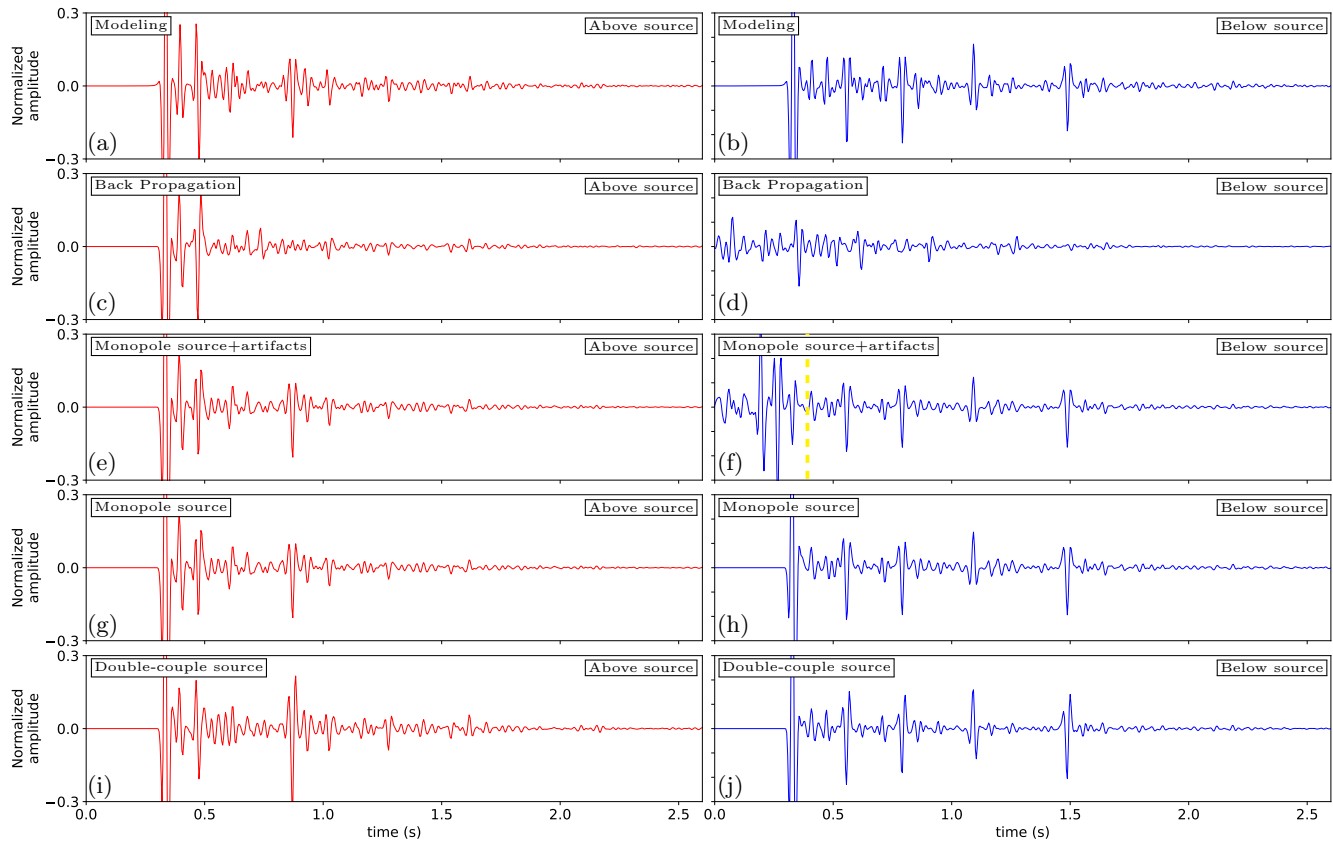

**Figure 7.** Traces from receivers in the subsurface at two locations, extracted from Figure 6. In the left column, the receiver is located above the source and corresponds to the red dot in Figure 6-(a) and in the right column it is located below the source and corresponds to the blue dot in Figure 6-(a). (a)-(b) Directly modeled wavefield using the exact model from Figures 5-(a) and (b). (c)-(d) Back-propagated wavefield obtained using equation (23). (e)-(f) Wavefield in the subsurface, retrieved for virtual receivers and a virtual source using equation (15). The yellow line in (f) indicates the time after which the correct signal is retrieved. (g)-(h) Similar as (e)-(f), for the homogeneous wavefield using equation (16). (i)-(j) Similar as (g)-(h), using equation (19) and a double couple signature inclined at an angle of 45 degrees. All wavefields have been convolved with a 30 Hz Ricker wavelet.

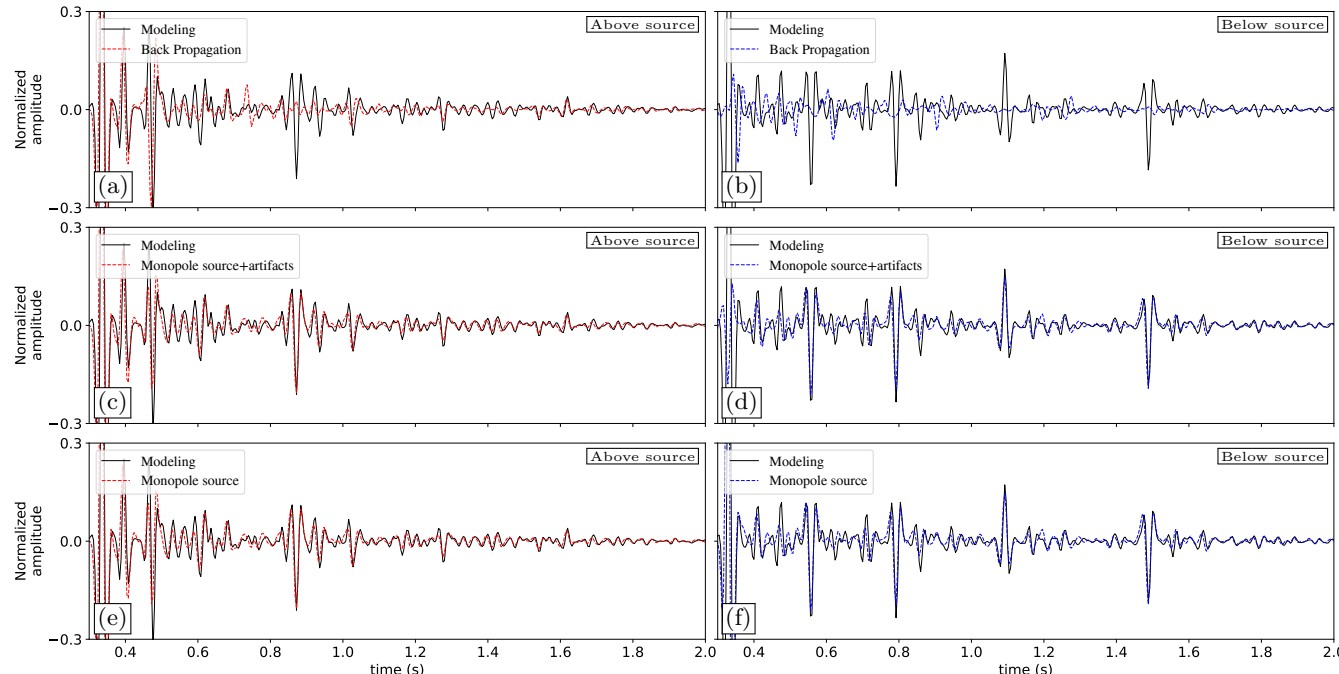

**Figure 8.** (a) Overlay of the traces from Figures 7-(a) and (c). (b) Similar as (a) for the traces from Figures 7-(b) and (d). (c) Similar as (a) for the traces from Figures 7-(a) and (e). (d) Similar as (a) for the traces from Figures 7-(b) and (f). (e) Similar as (a) for the traces from Figures 7-(a) and (g). (f) Similar as (a) for the traces from Figures 7-(b) and (h). All wavefields have been convolved with a 30 Hz Ricker wavelet.

### 3.1.2 Double-couple sources along extended faults

Until now, we only considered single point sources that have a symmetric signal. To study the situation of induced seismicity, we simulate a source that evolves over time over a larger area than a single point. We achieve this by placing a collection of sources along a line in the model. For this purpose, we place 131 sources along the fault plane that was indicated in Figure 5, starting at the bottom left corner, with a spacing of 7.07 m. The time between the activation of the shots is 12 ms, simulating a propagation speed of the source along the fault of 589 $\frac{m}{s}$. The fault is inclined at 45 degrees, therefore we make use of double-couple sources that are inclined at the same angle. We consider two scenarios, one where we have virtual sources and one where we have a measurement of a real source.

For the first scenario, we approach the problem by considering each source position separately. We do this by retrieving the homogeneous wavefield for each virtual source location separately and by shifting and superposing the results, similar to equation (21). Causality is applied to each individual wavefield before the superposition to avoid overlap between the causal and acausal part of the wavefields. Snapshots of the results are shown in Figures 9-(a)-(d), for 0, 500, 1000 and 1500 ms. The reason for the large timesteps is to ensure that all the sources along the fault have been activated during the final snapshot. The propagation of the source location along the fault is clear in these snapshots, however, a propagating wavefield appears to

be largely absent, with only a few events and ringing effects present. The reason for this phenomenon is that the velocity at which the sources are activated along the faults is lower than the propagation velocity of the medium. This effectively means that the phase velocity of the combined wavefield along the fault is lower than the propagation velocity of the medium and the emitted wavefield therefore becomes evanescent. This effect can be seen more clearly by considering the traces from two receiver positions. Similar to Figure 7, we extract the same receiver locations to consider the individual traces, as shown in Figure 10. In Figures 10-(a)-(b), the trace for the receiver location above the shallowest source location shows a trace with few events, except for some high amplitude events. The receiver location below the deepest source shows a trace that contains more ringing effects with a uniform amplitude. Because the amplitudes are similar and the events located close together, little information can be gained from this trace.

In reality, faults are not uniform, rather they are strongly heterogeneous, which causes variations for the source amplitude along the fault plane. To account for this effect, we apply random scaling to each source location along the fault plane before the superposition takes place. Applying a random scaling factor to the wavefield only affects the amplitudes of the wavefields and does not affect the arrival times or presence of the events in the wavefield. The result of this approach is shown in Figures 9-(e)-(h). The propagation of the source location along the fault is similar to the uniform amplitude approach, however, the individual wavefields are visible due to the random amplitude approach. Both the first arrivals and the codas can be seen, although there is much overlap between all the wavefields which makes distinguishing individual events at later times challenging. When the two receiver traces in Figures 10-(c)-(d) are studied, this challenge is still present. The trace contains events, however, it is difficult to say whether these events correspond to the response of one source or another.

To make an estimation for the arrival times of the retrieved response, we numerically model the line source in the subsurface, using the same random amplitude distribution as in the previous case. As we lack the capability to model snapshots of the response to the double-couple source acoustically, we make use of monopole point sources, instead of double-couple sources. As a result, the amplitudes of the events should not be compared to the retrieved response, however, the arrival times can be compared. The wavefield in Figures 9-(i)-(l) shows that the arrival times are well comparable between the modeling result and the retrieved response. This is further proven when the traces in Figures 10-(e)-(f) are considered. The arrival times have a strong match, while the amplitudes are not comparable. This confirms that the correct events are retrieved through this approach.

Next, we consider a different scenario, with a real source instead of a virtual one. Here, we once again retrieve the wavefield response of each source separately. However, instead of retrieving a separate wavefield for each of these responses and then superposing these results together, we superpose the responses before the wavefield is retrieved, following equation (22). By using this approach we obtain a response record that matches the response of a real source recording in the subsurface. The same random amplitude distribution that we used for the previous two results is applied for this approach as well, to make the comparison fair. The wavefield that is obtained is shown in Figures 9-(m)-(p), where we can see that the propagation of the source location along the fault is captured properly. There are issues with the approach due to the limitation of the representation that is used. The response to each source has artifacts that arrive before the first arrival when the virtual receiver is located below any of the source locations. These effects overlap with the causal wavefields of sources at other locations, and

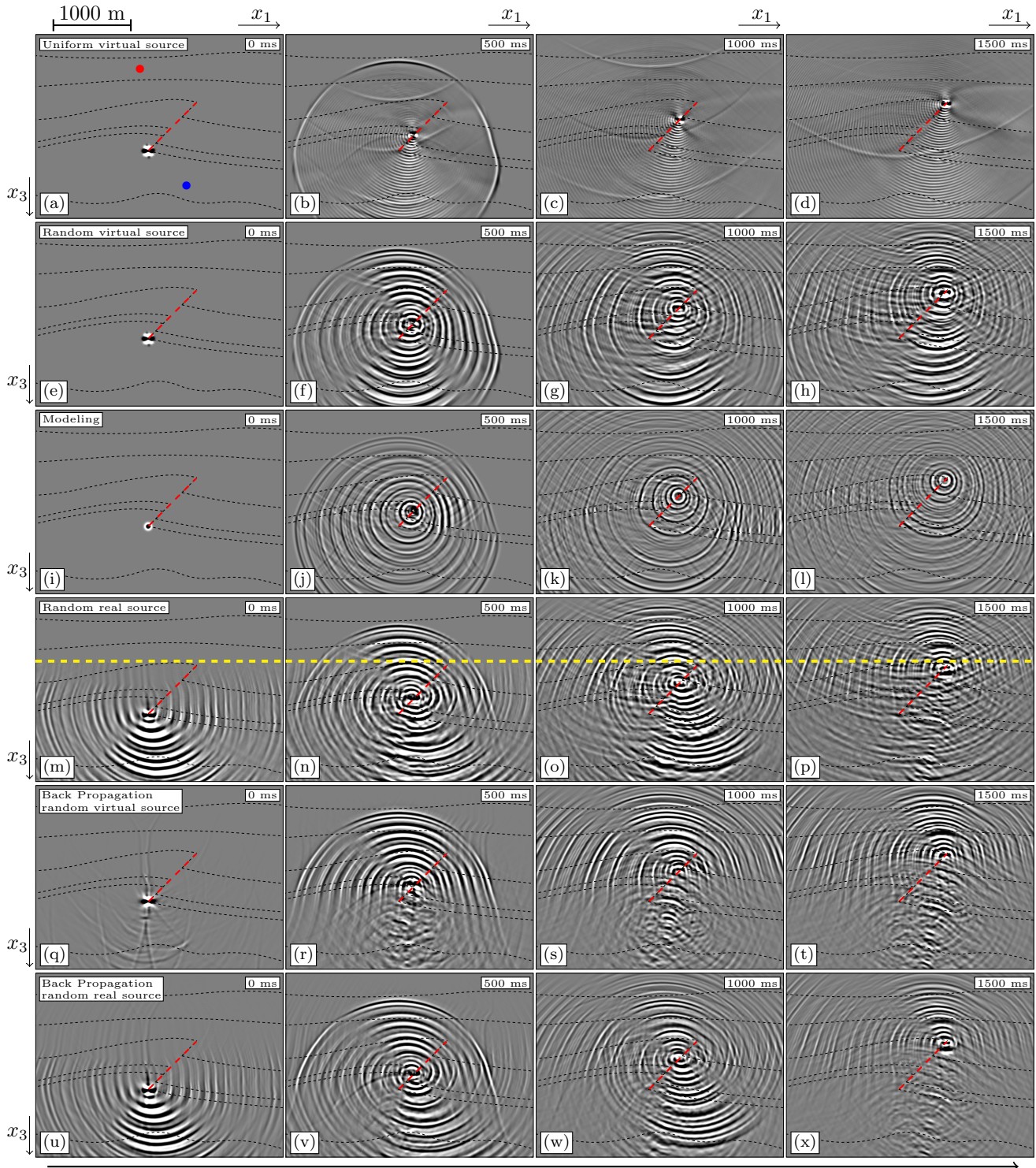

**Figure 9.** Snapshots of the wavefield inside the white box in Figure 5 for line sources. (a)-(d) Response in the subsurface, retrieved using equation (21) for virtual receivers and virtual double-couple sources inclined at 45 degrees with an uniform amplitude. (e)-(h) Similar as (a)-(d), using random amplitudes for the source. (i)-(l) Directly modeled wavefield using the exact model from Figures 5-(a) and (b) and monopole point sources with a random amplitude. (m)-(p) Similar as (e)-(h) using a superposition of double-couple sources with random amplitudes using equation (22). The yellow line indicates the border between the area below and above the shallowest source. (q)-(t) Similar as (e)-(h), however instead of using the homogeneous Green's function retrieval, the back propagation using equation (23) is used for each source position. (u)-(x) Similar as (m)-(p), however instead of using the Green's function retrieval, the back propagation using equation (23) is used. All wavefields have been convolved with a 30 Hz Ricker wavelet. The red and blue dot indicate the locations of the traces in Figure 10. The black dashed lines indicate the locations of geological layer contrasts.

obscure the events that should be present. Additionally, the downgoing first arrival is missing for all source locations. These problems are inherent to the representation and cannot be easily avoided, however, the coda of the response for later times will be correct, as we saw already for the point source in Figures 7-(e)-(h). When the traces for this approach from Figures 10-(g)-(h) are studied, we can see that if the receiver is located below the source locations, individual events belonging to the sources are impossible to distinguish. If the receiver is located above all the sources, however, the response is retrieved correctly. The lower receiver contains the correct coda at later time. We indicated this moment with a yellow line in Figure 10-(h), similar to Figure 7-(d). This, combined with the fact that the source location of the signal can be clearly distinguished, shows that this approach has potential for field recordings.

Finally, as an example for the improvement of this approach over conventional methods, we repeat the retrieval of a fault plane source using the back propagation method. We consider both the approach for retrieving a virtual source and retrieving a real source. For the first approach, we retrieve the response for each source location, mute the acausal part of the response and shift it in time, to create one source signal. However, instead of using homogeneous Green's function retrieval to obtain the responses, we employ the classic back propagation and show the resulting wavefield in Figures 9-(q)-(t). While the primay upgoing wavefield is still captured, the coda and the downgoing wavefield are absent. Aside from the missing events, artifacts are present at all times in the result. When the extracted traces are considered in Figures 10-(i)-(j), we can see that the trace is completely different to the traces in Figures 10-(c)-(d). Due to the fact that the missing events and the artifacts shift along with the source position, it masks the entire trace.

The effects of the classical back propagation approach have a similar result when we repeat the experiment for our real source example. We use classical back propagation instead of Green's function retrieval on the simulated real source response and show the result in Figures 9-(u)-(x). Similar problems with the coda and the downgoing wavefield are present and the artifacts in the wavefield are still ocurring. The extracted trace above the source locations in Figure 10-(k) shows the same result as in Figure 10-(i), which is consistent with the previous results. The extracted trace below the source locations in Figure 10-(l) shows the strong degradation in quality and has no match with the desired result in Figure 10-(d). This shows that for both types of sources, real or virtual, the single-sided approach with a focusing function is an improvement over the classical approach using back propagation. Therefore, the latter approach will not be used for the field data.

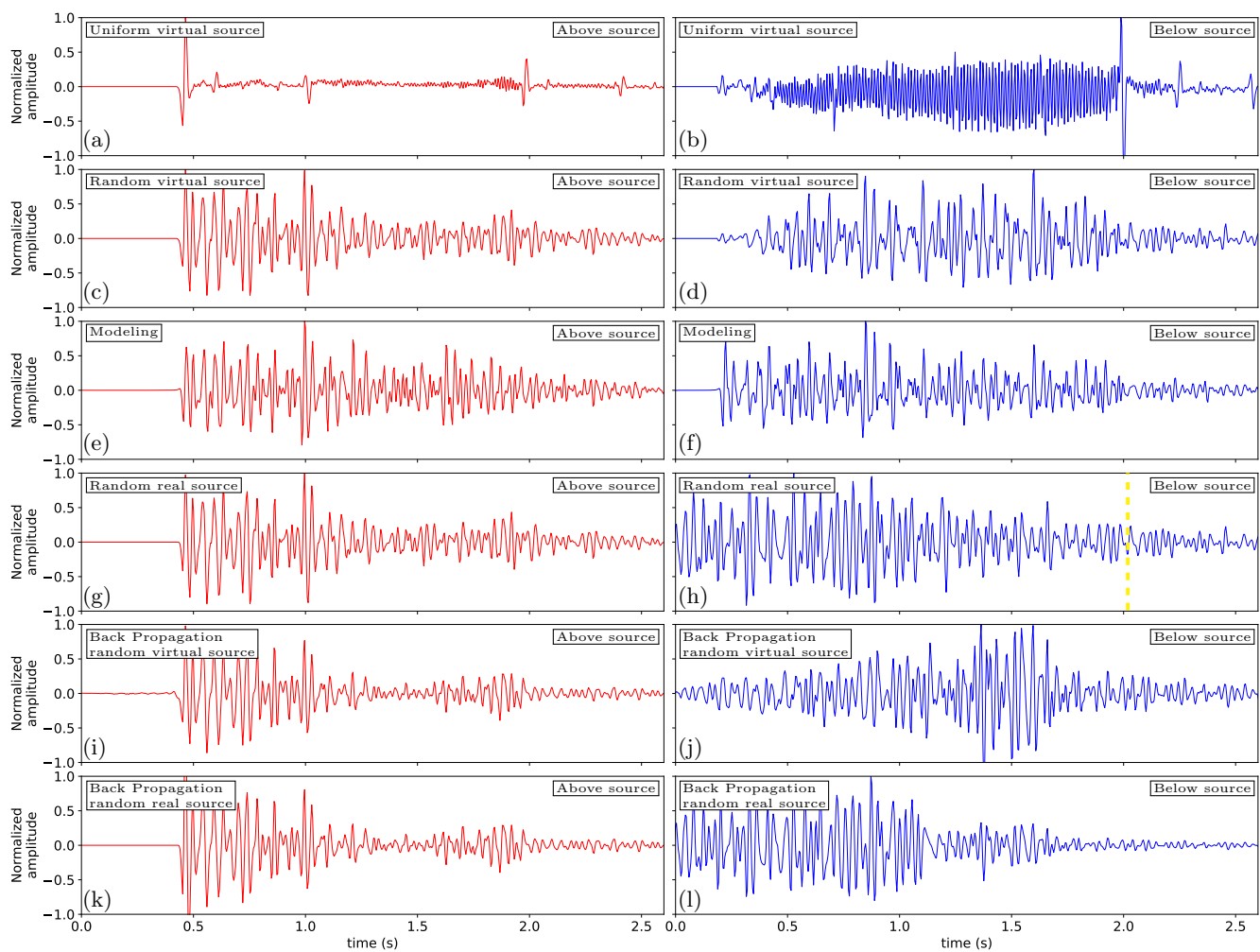

**Figure 10.** Traces of receivers in the subsurface at two locations, extracted from Figure 9. In the left column, the receiver is located above the source and corresponds to the red dot in Figure 9-(a) and in the right column it is located below the source and corresponds to the blue dot in Figure 9-(a). (a)-(b) Response in the subsurface, retrieved using equation (21) for virtual receivers and virtual double-couple sources inclined at 45 degrees with an uniform amplitude. (c)-(d) Similar as (a)-(b), using random amplitudes for the source. (e)-(f) Directly modeled wavefield using the exact model from Figures 5-(a) and (b) and monopole point sources with a random amplitude. (g)-(h) Similar as (c)-(d) using a superposition of double-couple sources with random amplitudes using equation (22). The yellow line in (h) indicates the time after which the correct signal is retrieved. (i)-(j) Similar as (c)-(d), however instead of using the homogeneous Green's function retrieval, the back propagation using equation (23) is used for each source position. (k)-(l) Similar as (g)-(h), however instead of using the Green's function retrieval, the back propagation using equation (23) is used. All wavefields have been convolved with a 30 Hz Ricker wavelet.

  **3.2   Field data results**

To demonstrate that our approach is not limited to synthetic data, we also apply the method on field reflection data. The field data were recorded in the Vøring basin, in a marine setting by SAGA Petroleum A.S., which is currently part of Equinor. Due to the setting, the receivers only recorded P-waves. The data consist of 399 common-source records, an example of which is shown in Figure 11-(c). The data were preprocessed before the application of the homogeneous Green's function retrieval,

 through the use of the Estimation of Primaries through Sparse Inversion (EPSI) method to remove the source wavelet, retrieve the near-offsets and remove the free-surface multiples (van Groenestijn and Verschuur, 2009). Moreover, we applied source-receiver reciprocity to allow the retrieval of two directions of offset and adaptive corrections to compensate for attenuation and incorrect source strength. Along with the reflection data, a smooth P-wave velocity model was also provided, which is shown in Figure 11-(a). We indicate the region of interest, where we will perform homogeneous Green's function retrieval, with a

 white dashed box. The model is not displayed in a true to life aspect ratio. The reflection data and the velocity model are the only inputs that are available for the homogeneous Green's function retrieval. No direct information about the subsurface is available for this area, however, using the reflection data and the velocity model, an image of the subsurface was created using the Marchenko method, shown in Figure 11-(b), which we will use as a reference for where scattering is expected to take place. This imaging was done independently of the homogeneous Green's function retrieval and is only used as a reference.

 More information about imaging using the Marchenko method, as well as an application on field data, can be found in Staring et al. (2018). The homogeneous Green's function retrieval for this dataset has been succesfully performed, as was shown in (Brackenhoff et al., 2019), however, in this work we will expand the results to include the line source configuration.

Because there is no information about the subsurface available, we cannot directly model in the subsurface and therefore have no benchmark, however, we have shown with the previous examples that the method is capable of retrieving the correct result.

 We perform homogeneous Green's function retrieval in the subsurface for both a virtual source and virtual receivers. The virtual source is a double-couple source, inclined at 20 degrees. The result is shown in Figures 12-(a)-(d) for 0, 300, 600 and 900 ms. The image of the subsurface from Figure 11-(b) is used as an overlay to help indicate the region where scattering of the wavefield is expected. The scattering takes place along regions where high amplitudes are present for the subsurface image, which indicates a match between the image and the homogeneous wavefield. Aside from the direct arrival, there is also a coda present, which contains several events. The result is not as clean as the synthetic data, however. This is due to the limitations of the field data. The data is attenuated, a problem that the Marchenko method cannot properly account for. The attenuation has been corrected for during the processing, however, this process is imperfect and will leave imperfections in the final result. There is also incoherent noise present in the field data, which has not been removed during the processing and will be present

 in the final result.

Figure 12-(a) shows a red and blue dot, which indicate the location of traces that are extracted and are shown in the left and right column of Figure 13, respectively. No benchmark for these traces is available, and thus it cannot be directly validated. The results in Figures 13-(a)-(b) show that the traces contain multiple well defined events, and that the noise on the trace is has lower amplitudes than these events. The amplitude of the first arrival is strong compared to the coda and the phase of all the

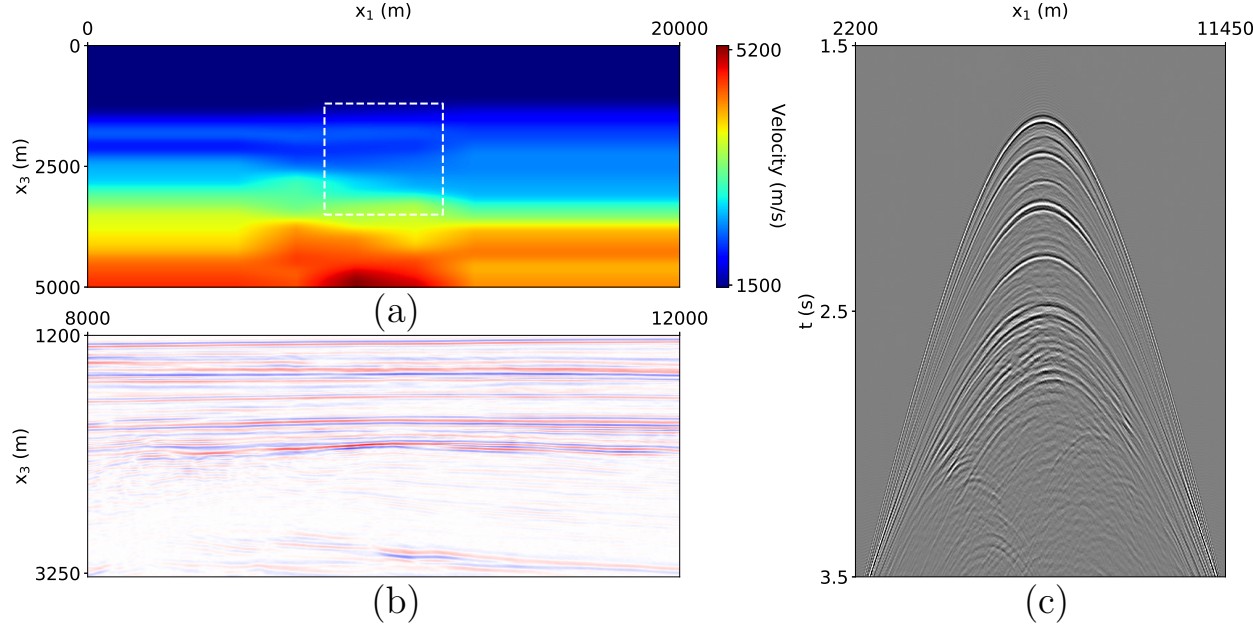

**Figure 11.** Real data example, (a) P-wave velocity in $\frac{\mathrm{m}}{\mathrm{s}}$ of the field data. The white box denotes the area of interest for the purpose of homogeneous Green's function retrieval. (b) Image of the subsurface located in the region indicated by the white dashed box. (c) Common-source record of the field reflection data, processed for the purpose of applying the Marchenko method. The reflection data source wavelet was reshaped to a 30 Hz Ricker wavelet. The data itself was recorded in the Vøring basin in Norway and was provided by Equinor.

events is similar. This shows that if the faults in the model are small compared to the wavelength, this approach can be useful for interpretation and characterisation of the source mechanism.

Next, we consider the two line source configurations for the virtual and the real source configuration. As there is no clear fault present in the model, the fault line is arbitrarily placed in the center of the model, inclined at an angle of 22.4 degrees. 161 sources are used with a spacing of 6.99 m, where the time between the activation of the shots is 12 ms, simulating a propagation speed of the source along the fault of 583 $\frac{\mathrm{m}}{\mathrm{s}}$. A random amplitude is assigned to each of the source locations to generate propagating waves. The first situation we consider is using equation (21), where homogeneous Green's function retrieval is performed for each location separately and the results are superposed and causality is imposed. The results of this approach are shown in Figures 12-(e)-(h), for 0, 1000, 2000 and 3000 ms. Similar to the synthetic data, the movement of the source is well captured and the first arrival and the coda are present in the signal. Part of the wavefield is not present, which corresponds to high angles at deeper depths, which, as we explained before, are not present in the reflection response and can therefore not be reconstructed. The result has a similar quality as the single double-couple source in Figures 12-(a)-(d) and the results on the synthetic data Figure 9.

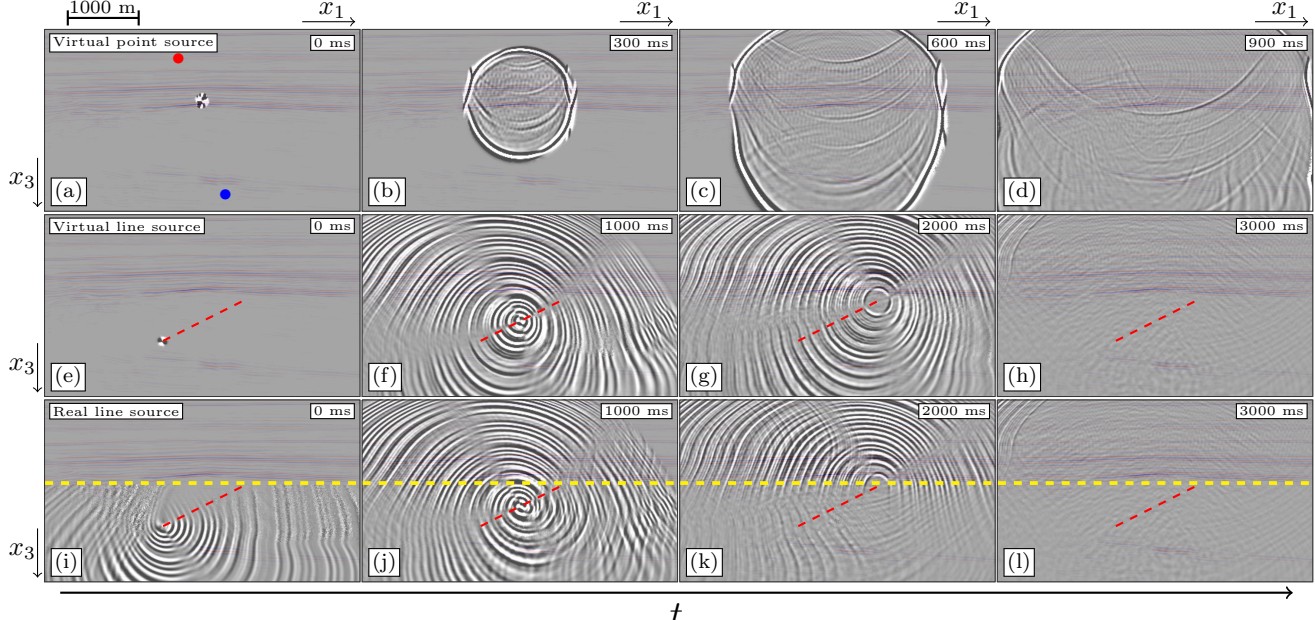

**Figure 12.** Snapshots of the wavefield inside the white box in Figure 11 for the field data. (a)-(d) Homogeneous wavefield in the subsurface, retrieved for virtual receivers and a virtual double-couple source inclined at -20 degrees using equation (19). (e)-(h) Similar as (a)-(d), for a line source of double couple sources with random amplitudes inclined at 22.4 degrees using equation (21). (i)-(l) Similar as (e)-(h), using a superposition of double-couple sources with random amplitudes using equation (22). The yellow line indicates the border between the area below and above the shallowest source. The images are overlain with the image of the subsurface from Figure 11-(b). All wavefields had their source wavelets reshaped to a 30 Hz Ricker wavelet.

There is no induced seismicity signal present for this area, so a real source signal cannot be used, but we simulate this as follows. Similar to the approach for the synthetic data, we use the Marchenko method to retrieve a wavefield response with a double-couple signature for each source location. These signals are then superposed to create a single source record, as a substitute for a real source signal. This approach follows equation (22), the results of which are shown in Figures 12-(i)-(l). Similar to the results for the synthetic data, the match between the two approaches above the shallowest source location is strong. This is proven further when the traces above the source from Figures 13-(c) and (e) are compared to each other. The traces are nearly identical. If we consider a location below the the deepest source location, the results are less comparable, again similar to the results that were achieved on the synthetic data. The traces for this location, shown in Figures 13-(d) and (f), support this conclusion. The match in this situation is non-existent for earlier times, and the information is hard to appraise. At later times, as indicated by the yellow line, the coda of the two approaches match each other, similar as seen before. For both types of retrieval, the source locations are well-defined in both time and space and not obscured by artifacts that could cast doubt on the source locations. Using both types of approach shows potential for the determination of the source location and the coda and can help in the characterisation of the fault mechanism.

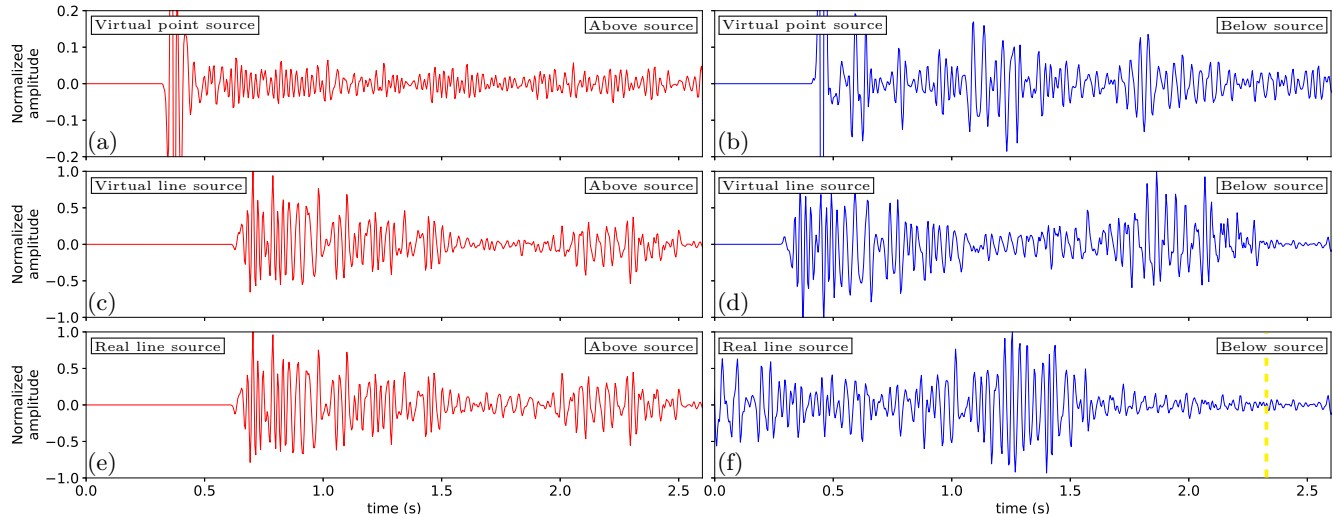

**Figure 13.** Traces of receivers in the subsurface at two locations, extracted from Figure 12. In the left column, the receiver is located above the source and corresponds to the red dot in Figure 12-(a) and in the right column it is located below the source and corresponds to the blue dot in Figure 12-(a). (a)-(b) Homogeneous wavefield in the subsurface, retrieved for virtual receivers and a virtual double-couple source inclined at -20 degrees using equation (19). (c)-(d) Similar as (a)-(b), for a line source of double couple sources with random amplitudes inclined at 67.6 degrees using equation (21). (e)-(f) Similar as (c)-(d), using a superposition of double-couple sources with random amplitudes using equation (22). The yellow line in (f) indicates the time after which the correct signal is retrieved. All wavefields had their source wavelets reshaped to a 30 Hz Ricker wavelet.

## 4 Conclusions

In this paper, we considered two methods to monitor full wavefields in the subsurface using the Marchenko method and found that in both cases, the Marchenko based approach is an improvement over classical methods such as back propagation. The first method is based on the creation of both virtual receivers and virtual sources in the subsurface. In this case, all the signals are created from the reflection data at the surface, and no response from a real subsurface source is used. For virtual point sources, we showed that we can assure that the source signal is symmetric and that therefore the full homogeneous wavefield can be retrieved without artifacts. The main limitation is that the steepest part of the wavefield at large depths cannot be retrieved. This approach works for virtual sources, both with a monopole signature and a more complex double-couple signature, the latter of which was used as a model for a small scale induced seismicity signal. Larger scale induced seismicity signals, emitted from a fault plane, were considered as well, simulated by a series of individual point sources with a double-couple signature. For this case, the homogeneous wavefield was retrieved for all the sources separately, after which the causal parts were isolated, shifted in time and superposed together. This produces a response from an extended fault rupture that is operating over a larger window of time, which produces a far more complex signal. All the source locations can be distinguished using this method. This method can be used to forecast in a data-driven way the response to possible future induced seismic events.

The second method we considered creates virtual receivers in the subsurface that observe a real response from a subsurface source. To this end, we considered point sources where the source signal was not assumed to be symmetric in time. The causal wavefield that is retrieved in this case is missing a part of the direct arrival and contains artifacts. These problems are only present when the virtual receiver is located below the source location, and the artifacts map exclusively in the time interval between the direct arrival of the wavefield and its time reversal. The coda of the causal wavefield is retrieved in full, as well as the source location of the subsurface response. When considering the responses propagating from a fault, the artifacts are more severe. Unlike in the method with the virtual sources, to simulate the response to a real rupturing fault, we shifted and superposed the source responses before the Green's function retrieval. Because of this, the artifacts are present for each point source, however, due to the time shift, the artifacts of one response coincided with the causal coda of other responses. As a result the coda of the retrieved wavefield is only partially obtained. The source locations of the fault response are retrieved correctly. This method can be used to monitor in a data-driven way the response to actual induced seismic events everywhere between the surface and the source.

We applied the two methods to synthetic and field data. For the synthetic data we showed that the retrieved responses match very well with directly modelled responses. The results obtained from the field data are very similar to those obtained from the synthetic data. The results on the datasets show the potential for the application of the method on real source signals in the future.

*Code availability.* The modeling and processing software that has been used to generate the numerical examples in this paper can be downloaded from https://github.com/JanThorbecke/OpenSource

*Data availability.* The seismic reflection data analysed in Figure 11, 12 and 13 are available from Equinor ASA, but restrictions apply to the availability of these data, which were used under license for the current study, and so are not publicly available. Data are however available from the authors upon reasonable request and with permission of Equinor ASA.

*Video supplement.* The videos of the snapshots of Figures 6, 9 and 12 can be found in https://github.com/JanThorbecke/OpenSource/tree/master/movies

*Author contributions.* JB wrote the paper. KW and JB devised the methodology. JB and JT developed software and generated the numerical examples. All authors reviewed the manuscript.

*Competing interests.* The authors declare that they have no competing interests.

*Acknowledgements.* This work has received funding from the European Union's Horizon 2020 research and innovation programme: European Research Council (grant agreement 742703).

The authors thank Equinor ASA for giving permission to use the vintage seismic reflection data of the Vøring Basin and Eric Verschuur and Jan-Willem Vrolijk for their assistance with the processing of the vintage seismic reflection data.

The authors thank Matteo Ravasi, Dominic Cummings and an anonymous reviewer for their review and suggestions to improve this paper.

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
