# Peer review of "Monitoring induced distributed double-couple sources using Marchenko-based virtual receivers"

_Solid Earth, 2018_

## Referee Comment (RC1) · Matteo Ravasi (Referee) · 24 Feb 2019

Dear Authors, I enjoyed reading your paper and I found the topic and content very relevant for the Solid Earth community. In your paper you explain a new approach to retrieval of full wavefield subsurface-to-subsurface responses which could be used to numerically reproduce the response of a microseismic event or earthquake at any point in the subsurface given its recording only at the Earth of the surface.

This is very appealing as per today locating microseismic events is mostly done only by taking into account the first (direct arrival) and by means of traveltime stacking (or backpropagation in a smooth model). While the first approach is very simple, it can

directly produce as output a map (or a volume) of the events in a area of interest, on the other hand the second approach leaves us with just a wavefield in time and space and subsequent processing (i.e., imaging condition) has to be carried out to produce what is really useful (a map or volume of the events). I find your approach an evolution, or perhaps improvement, of the latter when it comes to the creation of the subsurface wavefield (as also backpropagation is effectively creating a 'wrong' homogenous green's function from the unknown source location to any point in the modelling grid), in this view I feel that you are still lacking a second step to prove that having more detailed (but also complicated) wavefields is actually bringing some value in the localization of events - either accuracy or resolution or both. For this reason I believe that at least one of the numerical examples deserves special attention onto what can we actually do with these wavefields and perhaps using a standard IC for RTI and comparing its result with that produced by standard wavefield backpropagation may be enough to support your point. Otherwise I would fear that some readers may wonder if you are suggesting to monitor microseismicity by simply looking at wavefield snapshots without performing any post-processing to it.

Other than this point, I find the paper very well written, both the theory and examples are clear and easy to follow. I will be happy to reccomend your paper for publication as soon as you have addressed my main comment.

Best wishes MR

---

## Referee Comment (RC2) · Anonymous Referee #2 · 27 Feb 2019

The paper "Monitoring induced distributed double-couple sources using Marchenko-based virtual receivers" by Brackenhoff et al. proposes a method to create virtual receivers to monitor the response from subsurface sources. The paper is very well written and the underlying theory, which is mainly developed in a companion paper, is briefly laid out. The numerical examples using both synthetic and field data are well chosen and show nice applications of the proposed strategies. I appreciate that the authors make the source code available open source to reproduce the examples. I think this will be a nice paper that is relevant and interesting to the target audience.

The only major point that I find missing is a slightly more quantitative analysis of the

numerical results. You mainly focus on showing (normalized?) snapshots of the wave-fields and traces. For the synthetic tests, where you have the full modeled wavefield available, I would suggest to include plots of the differential wavefields as well as error plots. From the current figures I find it hard to judge the accuracy of the method.

I have a few more minor comments, which I list below. p.3, eq. (1): I was wondering if there is a particular reason for using a negative sign and the time derivative for the delta source?

p.3, l. 20: Typo: Missing closing parentheses .

p.4, Fig. 1: I know it is just a sketch, but I would recommend to add a colorbar for the velocities.

p. 5, l. 3: Instead of "We will not consider" I would rather say, "we will not describe / explain this method".

p. 5. l. 22: Typo: "an arbitrarily"

p. 5. l. 32: Instead of just "where functions" are available, I would explicitly mention what you are referring to. I assume Green's functions?

p. 6, Fig. 2: Is there a reason why you use the time-domain in the annotations, but the frequency domain in the representation in eq. (10)?

p. 6, l. 16: There is an extra space after reversal.

p. 9, Fig. 3: I am not sure if this figure is necessary. Is it just to show that the wavefields emitted by monopole and double-couple point sources are different? If you decide to keep the figure, I would suggest to at least change the caption and say "Sketch of the wavefields caused by . . ." instead of "Difference between".

p. 10, l. 5: No comma after superscript k.

p. 13, l. 16: Typo: "in" instead of "it".

[Figure]

p. 14, Fig. 5: I was wondering whether plotting the differential wavefield in (e) – (h) and in (i) – (l), respectively, would make it easier to see the differences? On a printout, the contrast between the wavefield and the background medium is pretty poor. Maybe they grayscale is not needed for the medium and/or you can plot the wavefield in color.

p. 15, Fig. 6: Please plot the errors between modelled and virtual receivers in addition to the absolute signals.

p. 16 l. 9 – 13: Could you comment to what extend the results are affected by the specific choice of the random scaling? For instance, would two seeds of random scaling factors still result in similar wavefields? As a related question: are you using the same seeds for the random amplitudes in Fig. 7 (e) – (h), and (i) – (l), respectively?

p. 18, l. 10 Could you please provide a few details, how the data was preprocessed?

p. 19/21, Fig. 9/10: The aspect ratio of the white box in Fig. 9(a) looks different than the zoom-in in Fig. 9(b) and Fig. 10.

p. 22, Fig. 11: Shouldn't the label in (e) be "Real line source" instead of "Virtual real source"?
* * *

---

## Author Response (AR1)

**1 General response**

We have noted the response to the two reviewers below and addressed their specific comments, however, we wish to add some additional comments. The authors have reviewed the document and made changes to some of the structure to improve the message for future readers. Two new figures, 3 and 8, have been added. All the other changes are indicated in the included document that tracks all the changes through the use of "latexdiff". Text or equations that has been removed is marked in red, while added or changed text is marked in blue. We hope that the responses and improvement to the paper are satisfactory to the editor and reviewers and we thank them for their time and effort.

**2 Response to Matteo Ravasi**

*Dear Authors, I enjoyed reading your paper and I found the topic and content veryrelevant for the Solid Earth community. In your paper you explain a new approach toretrieval of full wavefield subsurface-to-subsurface responses which could be used tonumerically reproduce the response of a microseismic event or earthquake at any pointin the subsurface given its recording only at the Earth of the surface.This is very appealing as per today locating microseismic events is mostly done onlyby taking into account the first (direct arrival) and by means of traveltime stacking (orbackpropagation in a smooth model). While the first approach is very simple, it can directly produce as output a map (or a volume) of the events in a area of interest, onthe other hand the second approach leaves us with just a wavefield in time and spaceand subsequent processing (i.e., imaging condition) has to be carried out to producewhat is really useful (a map or volume of the events). I find your approach an evolution,or perhaps improvement, of the latter when it comes to the creation of the subsur-face wavefield (as also backpropagation is effectively creating a 'wrong' homogenousgreen's function from the unknown source location to any point in the modelling grid). I feel that you are still lacking a second step to prove that having more detailed(but also complicated) wavefields is actually bringing some value in the localization ofevents - either accuracy or resolution or both. For this reason I believe that at least oneof the numerical examples deserves special attention onto what can we actually dowith these wavefields and perhaps using a standard IC for RTI and comparing its resultwith that produced by standard wavefield backpropagation may be enough to supportyour point. Otherwise I would fear that some readers may wonder if you are suggestingto monitor microseismicity by simply looking at wavefield snapshots without performingany post-processing to it.Other than this point, I find the paper very well written, both the theory and examplesare clear and easy to follow. I will be happy to reccomend your paper for publication assoon as you have addressed my main comment. Best wishes MR*

We thank the reviewer for his highly positive review and his suggestion to improve this paper. We would like to address his only major comment here. To show the reviewer that the approach using the Marchenko method is an improvement over previous approaches such as back propagation and RTI, we have added the results from back propagation experiments to the paper as both snapshots and extracted traces. We have performed these experiments only for the synthetic data and not the field data, as the synthetic data clearly shows that the Marchenko approach is superior. We hope that this demonstrates to the reviewer that our approach is an improvement over the classical methods.

Additionally, we would like to emphasize that our method is not another source localization method. We have emphasized in the text that the method is intended to forecast, in a data-driven way, the full wavefield response to possible future induced seismicity events using the two-step process and to monitor the full wavefield of actual seismicity events in the subsurface using the one-step process.

**3    Response to anonymous reviewer**

*The paper "Monitoring induced distributed double-couple sources using Marchenko-based virtual receivers" by Brackenhoff et al. proposes a method to create virtualreceivers to monitor the response from subsurface sources. The paper is very well written and the underlying theory, which is mainly developed in a companion paper, is briefly laid out. The numerical examples using both synthetic and field data are wellchosen and show nice applications of the proposed strategies. I appreciate that the authors make the source code available open source to reproduce the examples. I think this will be a nice paper that is relevant and interesting to the target audience.The only major point that I find missing is a slightly more quantitative analysis of the numerical results. You mainly focus on showing (normalized?) snapshots of the wave-fields and traces. For the synthetic tests, where you have the full modeled wavefieldavailable, I would suggest to include plots of the differential wavefields as well as errorplots. From the current figures I find it hard to judge the accuracy of the method.* We thank the anonymous reviewer for their comments and suggestions to help improve this paper. Changes to this article have been tracked through the use of "latexdiff" to demonstrate our revisions to the article. We have some more specific responses to some of the comments.

Regarding the reviewers comments about the accuracy of the method, in order to obtain the first arrivals, we make use of a smoothed version of the model without any density information, this is to simulate a very realistic version for the retrieval of the focusing functions. In a realistic setting, this is the type of data that will be available. Because of these limitations, the exact amplitude and the sampling of the wavefields are not exactly one to one with a directly modeled wavefield, making a quantitative comparison between the modeled wavefield and the retrieved wavefield very difficult. As such, we used the extracted traces of the data for a more detailed comparison.

To improve the comparison, we have added a figure that contains a zoom in of the modeled and retrieved traces overlying each other (Figure 8). This shows some of the sampling issues and we have added a discussion of these results in the paper. We hope that this helps to assess the accuracy concern about the paper.

As to the minor comments, we have addressed them individually as seen below:

*p.3, eq. (1): I was wondering if there is a particular reason for using a negative sign and the time derivative for the delta source?*

In the supplement the wave equation that is employed is derived. Because the source term is located in the stress-strain relation and this equation needs to be subtracted from the equation of motion the negative sign is introduced. The derivative is used to simulate a volume injection rate source rather than a pure point source.

*p.3, l. 20: Typo: Missing closing parentheses*

We added the closing parentheses. See the marked up version of the document.

*p.4, Fig. 1: I know it is just a sketch, but I would recommend to add a colorbar for the velocities.*

The image is a schematic, high contrast representation of the medium and does not represent the actual velocities. As such, adding a colorbar would not make sense. The image has therefore been reconstructed using the actual velocities and a colorbar has been added.

*p. 5, l. 3: Instead of "We will not consider" I would rather say, "we will not describe /explain this method".*

We changed the wording to "We will not explain this method". See the marked up version of the document.

*p. 5. l. 22: Typo: "an arbitrarily"*

We fixed the typo. See the marked up version of the document.

*p. 5. l. 32: Instead of just "where functions" are available, I would explicitly mention what you are referring to. I assume Green's functions?*

This part is indeed unclear. We mean in this case the receiver locations of the focusing functions and Green's functions. This has been added to the document.

*p. 6, Fig. 2: Is there a reason why you use the time-domain in the annotations, but the frequency domain in the representation in eq. (10)?*

The reason is that these type of data that are measured and retrieved through the Marchenko method will be available in the time-domain. The application that we use in the form of eq. (10) makes use of the frequency domain versions. To avoid confusion however, we have changed the quantities to the frequency domain in the figure.

*p. 6, l. 16: There is an extra space after reversal.*

We have removed the space. See the marked up version of the document.

*p. 9, Fig. 3: I am not sure if this figure is necessary. Is it just to show that the wavefields emitted by monopole and double-couple point sources are different? If you decide to keep the figure, I would suggest to at least change the caption and say "Sketch of the wavefields caused by..." instead of "Difference between".*

5    We have decided to keep the figure, but we have added the suggested change by the reviewer. See the marked up version of the document.

*p. 10, l. 5: No comma after superscript k.*

The comma has been removed. See the marked up version of the document.

*p. 13, l. 16: Typo: "in" instead of "it".*

10    The typo has been fixed. See the marked up version of the document.

*p. 14, Fig. 5: I was wondering whether plotting the differential wavefield in (e) – (h)and in (i) – (l), respectively, would make it easier to see the differences? On a printout,the contrast between the wavefield and the background medium is pretty poor. Maybe they grayscale is not needed for the medium and/or you can plot the wavefield in color.*

Please see the response we have posted to your general comments. About the reason for the grayscale, this paper is a companion 15    paper and across the papers, we have decided on a uniform style for the plotting of the wavefields. We have used another clipping factor to improve the visual.

*p. 15, Fig. 6: Please plot the errors between modelled and virtual receivers in addition to the absolute signals.*

Please see the response we have posted to your general comments.

*p. 16 l. 9 – 13: Could you comment to what extend the results are affected by the specific choice of the random scaling? For* 20    *instance, would two seeds of random scaling factors still result in similar wavefields? As a related question: are you using the same seeds for the random amplitudes in Fig. 7 (e) – (h), and (i) – (l), respectively?*

The scaling of the wavefield only affects the amplitude of the events and does not change the presence of events in the wavefield or their arrival times. We have added this to the document. On your related question, yes the same amplitude scaling is used, which is mentioned in the text of the document, however, to avoid confusion, we have made this clearer in the text.

*p. 18, l. 10 Could you please provide a few details, how the data was preprocessed?*

110    The data was processed through the use of EPSI, source-receiver reciprocity and adaptive corrections for attenuation and incorrect source strength. We have added these details to the document.

*p. 19/21, Fig. 9/10: The aspect ratio of the white box in Fig. 9(a) looks different than the zoom-in in Fig. 9(b) and Fig. 10.*

The aspect ratio of Fig. 9 (a) is not true to life, as the model is much longer in horizontal direction than in vertical direction. For aesthetic reasons, we have decided to plot the data like this, rather than true to life, however, the extent of the model has 115    been plotted. Figure 10 is plotted true to life to not distort the wavefields.

*p. 22, Fig. 11: Shouldn't the label in (e) be "Real line source" instead of "Virtual real source*

The label has been changed. See the marked up version of the document.

[revised manuscript text omitted]

---

## Referee Report (RR1)

Referee Report
Monitoring induced distributed double-couple sources using Marchenko-based virtual receivers
Joeri Brackenhoff, Jan Thorbecke, and Kees Wapenaar

Joeri, Jan, Kees, & the Editor:
The paper "Monitoring induced distributed double-couple sources using Marchenko-based virtual receivers" demonstrates a methodology for reconstructing the response to both monopole and dipole sources in the subsurface at virtual receiver locations throughout the medium using a Marchenko method, which is proposed to be used for forecasting the response throughout the medium to future induced seismic events, or for monitoring the response throughout the medium of actual seismic events that have already occurred.
I greatly enjoyed reading and reviewing this paper, and believe this work will be of great interest to the geophysics community. My recommendation to the editor is that this work be accepted subject to minor revisions, and I will separate my comments into the minor revisions necessary for this work to be published, and additional thoughts that I feel would make the paper easier to read. The following are all with respect to the main paper, as opposed to the supplementary materials.

Minor Revisions:

1. The choice of colour scheme in Figure 1, and as a consequence Figures 2 and 3, raises a number of issues. In Figure 1(a), the choice to overlay dark green arrows over a green background makes said arrows difficult to view on what is conceptually an excellent figure. A greater issue arises due to the overlaying of green arrows on a red background, which may be problematic for those with red-green colour-blindness. Given that the motif of red and green arrows to represent ray paths is a continued theme through Figures 2 and 3, I would suggest changing the colour map used to plot the underlying velocity model so as to highlight the arrows.

2. Page 19, line 11: *"The evanescent problem does not occur when the sources along the fault have random amplitudes"*. It is not obvious to me why this should be the case. Is this an axiom or a previously found result? If it is somebody's result then it needs an appropriate reference, if it is axiomatic then it would benefit from an explanation as to why this should be the case. You acknowledge 2 sentences after this statement that *"faults are extremely heterogeneous"*, if this forms part of the basis on which the previous statement was made then I believe this paragraph would benefit from minor restructuring of the opening 3 sentences to reduce any feeling of a circular logic being applied.

Additional thoughts:
1. Page 1, line 22: *"When the source is not active, but rather passive, such as when caused by an induced earthquake, the resulting signal can be measured as well"*. This sentence feels wordy and unconcise compared to the rest of the paper. Given that a source can be considered as either active or passive, and the previous description is explicitly about active source seismics, it doesn't seem necessary to state passive sources are not active. I would suggest something like

*"For passive sources, such as induced earthquakes, the resulting signal can be measured as well"*.

2. Page 4, line 15: *"For moderately inhomogeneous media".* Recognising that this type of statement is reasonably common in papers, this statement feels somewhat vague and imprecise compared to the majority of the paper. I'm not sure if there is a better way to express this however.

3. Figure 2: I believe that Figures 2(a) and 2(b) would benefit from each being slightly larger, as the current figures feel somewhat overcrowded with text labels and arrows, for example the "+" symbol in Figure 2(b) that is almost touching the arrow from $x_B$ to $x_A$. With a slightly larger figure there may also be space to use a heavier line width on the arrows, which I feel would also significantly improve the readability of the figure for the same reason as for Figure 1.

4. Figure 4: Whilst this figure serves its purpose of distinguishing the wavefield due to a monopole source from the wavefield due to a dipole source in the same medium, I immediately felt looking at this figure that placing the well-recognised "beach-ball" diagrams of such sources next to their respective wavefields might aid the reader in rapidly recognising the differences in the wavefields without needing to double-check for the changes in polarity around the wavefront. For example:

5. Figure 9: The green line does not show up clearly when printed on paper. I know that reading papers on a screen is increasingly popular, but there are some who will read it on paper and may miss this detail. A better contrasting shade of green or a wider line may help it stand out more.

6. Page 21, line 11: *"… we repeat the retrieval of the virtual source and the real source where we replace the retrieval of the wavefield by the classical back propagation".* As I understand it, you are saying that you repeat the virtual source methodology (dealing with each virtual source one by one), and the "real" source method (in this example it is a real synthetic source rather than a recording from an actual seismic survey as demonstrated in the following section) whereby the individual point sources are assumed to be inseparable and

thus processed as the summation of the sources, and apply the back-propagation method shown previously in Figures 6(e)-(h) for the monopole source case. I had to read over the preceding results a number of times before I decided what was going on. It would be unfair of the reader to ask the author to write everything in a manner that required no thinking at all, but if this were able to be expressed more clearly then it would be worthwhile to do so before publication.

7. Page 23, lines 18 & 19: *"More information about imaging using the Marchenko can be found …".* I presume that this is a typo and should say *"Marchenko method"*, in any case I would suggest changing this to something like *"More information about imaging of a field data example using Marchenko methods can be found…"* to emphasise the different approach used when processing field data as opposed to solely synthetic data examples.

---

## Author Response (AR2)

Response to referee report
by Dominic Cummings

We would like to thank the referee for his comments and suggestions and for taking the time to review our paper. In this response, we would like to answer his questions and suggestions. Most of the changes in the text can be found easily by checking the tracked changes version of our paper.

*Minor Revisions:*
*1. The choice of colour scheme in Figure 1, and as a consequence Figures 2 and 3, raises a number of issues. In Figure 1(a), the choice to overlay dark green arrows over a green background makes said arrows difficult to view on what is conceptually an excellent figure. A greater issue arises due to the overlaying of green arrows on a red background, which may be problematic for those with red-green colour-blindness. Given that the motif of red and green arrows to represent ray paths is a continued theme through Figures 2 and 3, I would suggest changing the colour map used to plot the underlying velocity model so as to highlight the arrows.*
We understand the reviewer's concerns. The colormap of the figure has been adjusted.

*2. Page 19, line 11: "The evanescent problem does not occur when the sources along the fault have random amplitudes". It is not obvious to me why this should be the case. Is this an axiom or a previously found result? If it is somebody's result then it needs an appropriate reference, if it is axiomatic then it would benefit from an explanation as to why this should be the case. You acknowledge 2 sentences after this statement that "faults are extremely heterogeneous", if this forms part of the basis on which the previous statement was made then I believe this paragraph would benefit from minor restructuring of the opening 3 sentences to reduce any feeling of a circular logic being applied.*
This was indeed poorly worded. We have changed this in the text, as can be seen in the track-changes version of the manuscript.

Additional thoughts:
*1. Page 1, line 22: "When the source is not active, but rather passive, such as when caused by an induced earthquake, the resulting signal can be measured as well". This sentence feels wordy and unconcise compared to the rest of the paper. Given that a source can be considered as either active or passive, and the previous description is explicitly about active source seismics, it doesn't seem necessary to state passive sources are not active. I would suggest something like "For passive sources, such as induced earthquakes, the resulting signal can be measured as well".*
We have made changes to the sentence, as can be seen in the track-changes version of the manuscript.

*2. Page 4, line 15: "For moderately inhomogeneous media". Recognising that this type of statement is reasonably common in papers, this statement feels somewhat vague and imprecise compared to the majority of the paper. I'm not sure if there is a better way to express this however.*
We find this the best expression for this type of situation, hence we have not adjusted it.

*3. Figure 2: I believe that Figures 2(a) and 2(b) would benefit from each being slightly larger, as the current figures feel somewhat overcrowded with text labels*

*and arrows, for example the "+" symbol in Figure 2(b) that is almost touching the arrow from x B to x A . With a slightly larger figure there may also be space to use a heavier line width on the arrows, which I feel would also significantly improve the readability of the figure for the same reason as for Figure 1.*

The figure has been restructured by changing the position of text and objects and the line thicknesses have been increased.

*4. Figure 4: Whilst this figure serves its purpose of distinguishing the wavefield due to a monopole source from the wavefield due to a dipole source in the same medium, I immediately felt looking at this figure that placing the well-recognised "beach-ball" diagrams of such sources next to their respective wavefields might aid the reader in rapidly recognising the differences in the wavefields without needing to double-check for the changes in polarity around the wavefront.*

A good suggestion, we have added the radiation patterns at the origins of the wavefields.

*5. Figure 9: The green line does not show up clearly when printed on paper. I know that reading papers on a screen is increasingly popular, but there are some who will read it on paper and may miss this detail. A better contrasting shade of green or a wider line may help it stand out more.*

The color of the green line has been adjusted to yellow. We made test prints ourselves to check the color and it appeared fine for us. We have made this change to the other figures of this type as well.

*6. Page 21, line 11: "... we repeat the retrieval of the virtual source and the real source where we replace the retrieval of the wavefield by the classical back propagation". As I understand it, you are saying that you repeat the virtual source methodology (dealing with each virtual source one by one), and the "real" source method (in this example it is a real synthetic source rather than a recording from an actual seismic survey as demonstrated in the following section) whereby the individual point sources are assumed to be inseparable andthus processed as the summation of the sources, and apply the back-propagation method shown previously in Figures 6(e)-(h) for the monopole source case. I had to read over the preceding results a number of times before I decided what was going on. It would be unfair of the reader to ask the author to write everything in a manner that required no thinking at all, but if this were able to be expressed more clearly then it would be worthwhile to do so before publication.*

We have made changes to the sentence to clarify our explanation, as can be seen in the track-changes version of the manuscript.

*7. Page 23, lines 18 & 19: "More information about imaging using the Marchenko can be found ...". I presume that this is a typo and should say "Marchenko method", in any case I would suggest changing this to something like "More information about imaging of a field data example using Marchenko methods can be found..." to emphasise the different approach used when processing field data as opposed to solely synthetic data examples.*

We have made changes to the sentence, as can be seen in the track-changes version of the manuscript.

[revised manuscript text omitted]